# Sandbag Replacement Systems - a nonsensical and costly alternative to sandbagging?

Lena Lankenau, Christopher Massolle, Bärbel Koppe, Veronique Krull

Institute for Hydraulic and Coastal Engineering, Hochschule Bremen – City University of Applied Sciences, Bremen, 28199, Germany

*Correspondence to*: Lena Lankenau (lena.lankenau@hs-bremen.de)

**Abstract.** In addition to the flood defence with sandbags, different sandbag replacement systems (SBRSs) have been available for a number of years for use in operational flood protection. The use of sandbags is time-consuming as well as highly intensive in terms of materials and personnel. In contrast, the use of SBRSs entails higher investment costs. However, SBRSs are reusable and require lower costs for helpers and logistics, so that the higher investment costs are offset by repeated use. So far, SBRSs are rarely used in Germany in operational flood protection. The reasons lie on the one hand in the different financing modalities of investment and operational costs and on the other hand in the low technical confidence in SBRSs. These problems are addressed by the research program of the Institute of Hydraulic Engineering at the Hochschule Bremen - City University of Applied Sciences (IWA). A series of systematic large-scale tests of sandbag systems and SBRSs with focus on functionality, stability and handling was carried out. It showed that the majority of the tested SBRSs are able to provide comparable protection as sandbag systems with a significantly reduced use of materials, logistics and helpers. Nevertheless, it is advisable to develop and perform well-defined certification tests for SBRSs, to define clear instructions for the use and to identify limits to the use of certain SBRSs. For example, not all systems work equally well on different surfaces.

Supplementary to the practical tests, costs for the procurement and use of various sandbag systems and SBRSs were determined on the basis of realistic scenarios. This will provide a methodology as well as concrete figures for the holistic costing for provision and use of different protection systems. It turned out that the higher investment costs for the investigated SBRSs compared to sandbag systems are already amortized on the second use of the reusable systems.

## 1 Introduction

The classic aid in operative flood defence is the sandbag. So-called sandbag replacement systems (SBRSs) have also been available for some time now, although their use is still very limited. Figure 1 shows such mobile, location-independent flood defence systems: they can be subdivided into tube, basin, flap, trestle, dam or panel systems and bulk elements. The systems counteract flooding either by their bulk weight, which is induced by water, sand, respectively concrete (container and bulk systems) or their geometry in connection with the vertical hydrostatic water pressure (flap, trestle, dam and air-filled tube systems – not shown in Figure 1), which either way results in frictional forces on the ground. Though, panel systems consist

of panels which are hold in place by sticks driven in the ground on alternate sides. But commonly, location-independent mobile flood protection systems do not need additional anchoring to the ground. However, some producers offer such a possibility, which introduces a safety surplus or can be necessary when high flow velocities or wind stress on the not jet impounded system are expected. Sandbags as well as SBRSs are used in flood disaster management – especially in case permanent flood protection

systems like dykes are failing or in case no permanent flood protection schemes are available because the currently endangered area was thought not to be at risk. Thus, sandbags as well as SBRSs are used in extreme flood events. There is no obligation to demonstrate the functionality of an SBRS so far. In general, however, SBRSs are suitable for flood protection and can be equated with sandbagging in terms of functionality (c.f. Pinkard et al. (2007), Niedersaechsischer Landtag (2014), Massolle et al. 2018). Although, depending on construction, geometry and filling of the individual system their safety against failure might

differ. Nevertheless, decision makers need to have reliable information about the functionality of an individual SBRS. This information is not always available, especially not from an independent source.

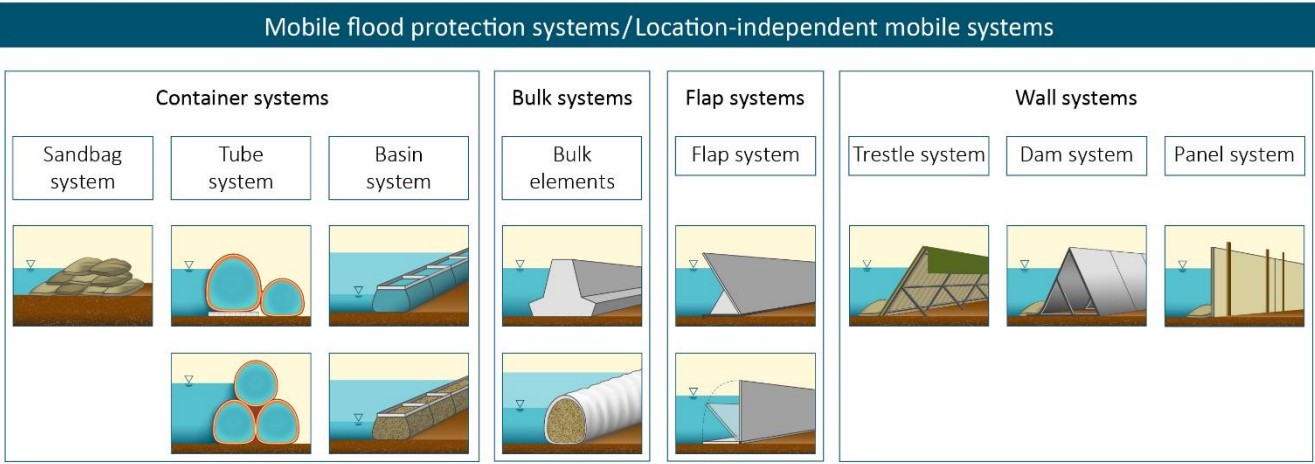

**Figure 1: Classification of mobile, non-location dependent flood protection systems (Massolle et al., 2018).**

Sandbagging is time-consuming as well as highly intensive in respect of materials and personnel. SBRSs in contrary, hold the

15 potential for a much more efficient flood defence, as their use entails significantly lower material, personnel and time requirements than conventional sandbagging. For example, 16 500 sandbags and 250 t of sand are required to build up a 100 m long and 1.0 m high sandbag dam (cf. THW, 2017). 60 helpers would need about 10 hours (cf. THW, 2017) only to fill the sandbags and set up the dam and yet the efforts for e.g. logistics of materials and supply of helpers are not considered. However, the advantage of using sandbags lies in the possibilities for flexible deployment and many years of practical experience. Figure

2 shows firemen raising a dyke by setting up a temporary sandbag dam.

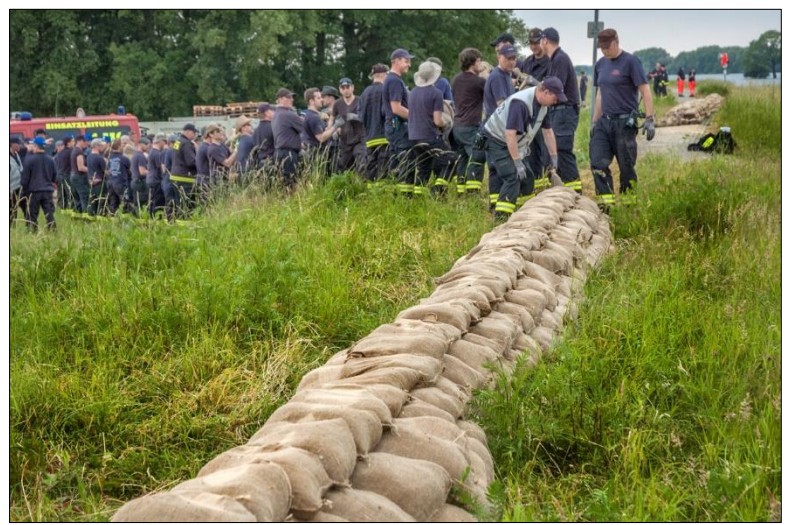

**Figure 2: Firemen during the Elbe flood in 2013. Setting up a sandbag dam to raise a dyke.**

SBRSs either do not need a filling at all or the filling respectively the systems are put in place with technical assistance such as pumps (water filling), wheel loaders (sand filling) or cranes (bulk elements made of concrete). Thus, the systems can be set
up and dismantled with considerably less time and manpower (cf. Massolle et al. 2018). Logistical efforts are minimized if no filling is needed or water, which can usually be obtained locally, is used. In contrast to sandbags, SBRSs are reusable and do not have to be disposed of at high cost after a flood event. From these points of views, SBRSs can also be suitable for scheduled flood protection measures in areas where no permanent flood protection schemes can be applied. The main disadvantage of SBRSs is the higher cost of acquisition. However, the lower expenditure on helpers, logistics and disposal of material means
that these higher investment costs can be offset through a reuse. Furthermore, there is limited confidence and a lack of knowledge in the functionality of SBRSs. Besides the non-confidence in the general functionality of a SBRS, fear of vandalism or mechanical influences e.g. impacts of flotsam or vehicles as well as the collective failure (domino effect) of a SBRS are of great concerns. In general, the functionality of sandbag dams can also be endangered by vandalism or mechanical influences but rather less by a collective failure, unless the sandbag dam is heavily overflowed over long distances.
Temporary flood dams made out of sandbags or linear SBRSs are set up in order to protect the hinterland from inundation. Beyond that sandbags are also used at the inner embankment securing saturated dykes either on selective points where there is considerable seepage (temporary ring dam) or over a larger area (load drain). Flutschutz offers corresponding SBRSs (Figure 3). For an explanation of the hydraulic situation at saturated dykes during a flood event see e.g. Simm et al. (2013). Sandbag dams and linear SBRSs are directly exposed to flooding. In contrary temporary ring dams and load drains are generally exposed
to lower loads as they are not subjected to the direct influence of high hydrostatic pressures or the dynamic impact caused by waves and flotsam. They are therefore less endangered in their functionality.

**Figure 3: Dyke defence measures for a saturated dyke over an extensive are (load drain) and for heavy punctual exit of seepage (temporary ring dam). Sandbagging (left) and corresponding SBRSs (right).**

In Germany, operational flood defence is regulated as part of hazard prevention or disaster control at the federal state level. Direct responsibility lies at the municipal level and thus with the local districts and cities. This includes the responsibility to provide the necessary material for the protection of the general public, whereby as a rule sandbags — which are the significantly cheaper option — are preferred over SBRSs. In case of a disaster event, assistance can be requested from the federal state or the federal government, whereby the financing of such assistance will still remain initially with the affected administrative districts or cities. Ultimately, the costs of major damage events, such as caused by the Elbe floods of 2002, 2006 and 2013, will be borne predominantly by the federal state and the federal government. Once such an event occurs, however, no time can be lost in procuring SBRSs, if they are not already standing by. Thus, the cost of procuring and stocking SBRSs—in addition to a lack of confidence or knowledge about their functionality—presents a major hurdle to their use.

Therefore, in Germany during the Elbe flood in 2013, SBRSs were only used in isolated cases (cf. AQUARIWA, 2019; Mobildeich, 2019), despite the fact that the use of sandbagging for operational flood defence is very time, material and labour intensive. Figure 4 shows two SBRSs after the Elbe flood in 2013. The two systems were successfully used to prevent the hinterland from flooding (Niedersaechsischer Landtag, 2014).

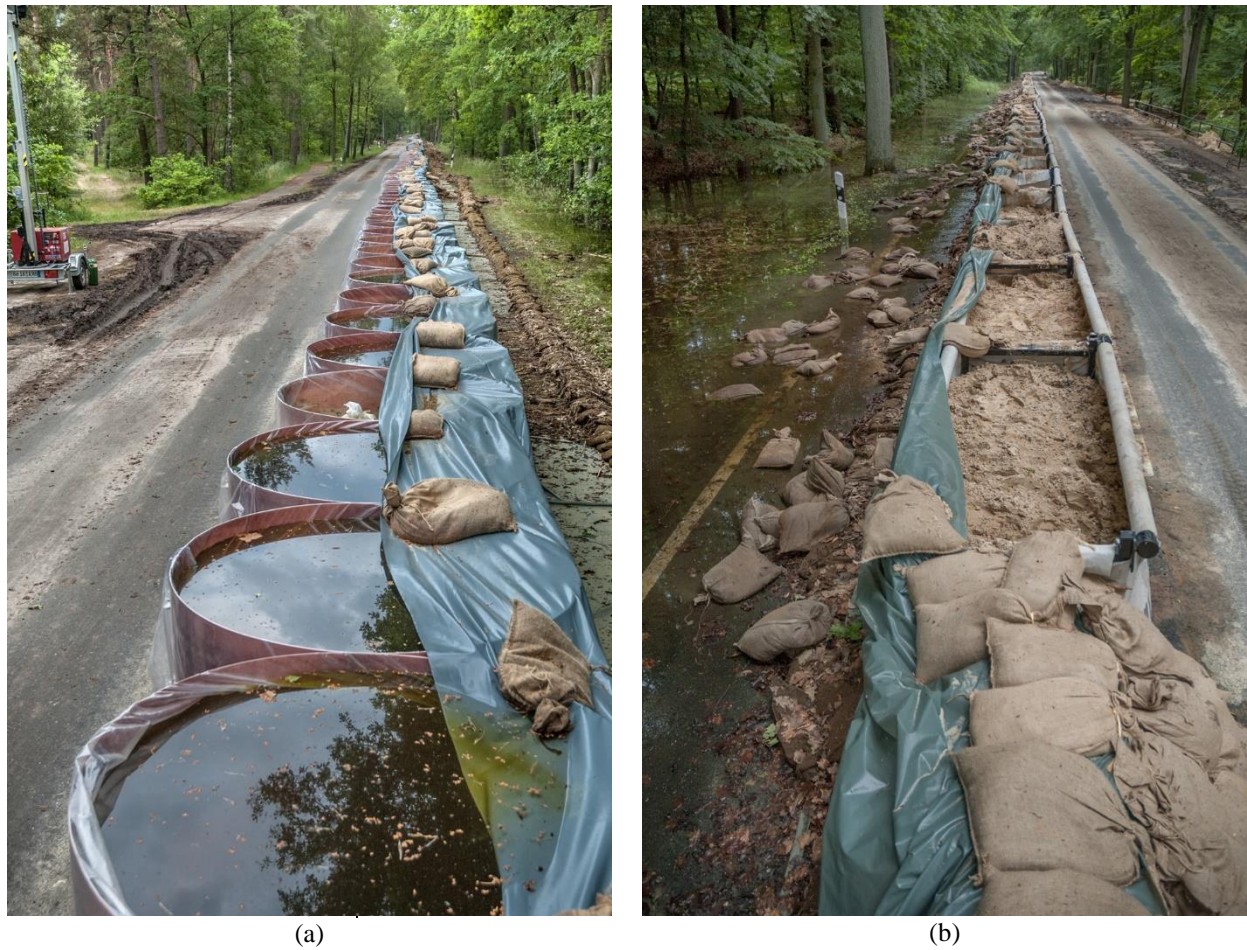

|     |     |
| :-: | :-: |
| (a) | (b) |

**Figure 4: SBRSs near Gartow (Lower Saxony, Germany) after the Elbe flood in 2013. (a) AQUARIWA, (b) Quick Damm type E.**

In order to increase the confidence of decision-makers in SBRSs and to promote the availability of only well-functioning SBRSs, it is desirable to carry out systematic tests on functioning, stability and handling and to develop relevant certification procedures. In addition to the functionality of SBRSs their costs and efficiency in terms of personnel, time and logistics compared to sandbagging should be investigated to likewise support decision-makers.

At the international level, corresponding certification already exists. It can be awarded by the globally active testing and certification service FM Approvals (FM Approvals 2019), based on the American National Standard for Flood Abatement Equipment (ANSI and FM Approvals, 2014), and the British Standard Institution (BSI, 2019a), which is based on the Publicly Available Specification (PAS) for flood protection products—Specifications Part 2: Temporary Products (BSI, 2014). Specific SBRSs certified by FM Approvals can be found under NFBTCP (2019) and SBRSs certified by BSI Kitemark can be found under BSI (2019b). In Germany, no corresponding certification or testing system for SBRSs is currently available. However, some information can be found on the design and the scheduled as well as unscheduled use of SBRSs in German-speaking countries, especially in the recommendations of the leaflet 'Mobile Flood Defence Systems' issued by the German Association

of Engineers for Water Management, Waste Management and Cultural Construction. (BWK, 2005), in the handbook 'Mobile Flood Protection' of the Austrian Water and Waste Management Association (ÖWAV, 2013) and in the decision-making aid 'Mobile Flood Protection' of the Swiss Association of Cantonal Fire Insurers (VKF) as well as the Swiss Federal Office for Water and Geology (BWG) (Egli, 2004).

There are relatively few publications on comparative studies of sandbagging and SBRSs. Within the scope of test setups in the test basin of the U.S. Army Corps of Engineers (USACE), one sandbag dam as well as two sand-filled container systems and one trestle system were investigated (Pinkard et al., 2007). In addition to the time spent on system installation and dismantling, the operational costs for a system set-up with a length of around 305 m and a height of around 0.91 m were also estimated. However, logistical aspects were not taken into account, and it was assumed that labour on the construction of the sandbag

dam would be on a free and voluntary basis. In addition, the sandbag requirement estimated in the study differs from the usual approaches in Germany, as the sandbag dam in the U.S. is constructed on a broader basis.

Investigations of the functionality of SBRSs were also carried out by the UK Environment Agency (EA) (Ogunyoye et al., 2011) on the basis of three sources of information; namely, the literature, user workshops and interviews with manufacturers and distributors of products. It was found that most of the systems provided adequate protection, but that in some cases

operational processes or inaccurate hydraulic assessments led to system failure. The assessments covered the physical, operational and structural characteristics of temporary flood products available on the UK market in 2009. The systems were subdivided into tubular systems, containers, freestanding barriers and frame barriers. The report furthermore highlights the relevance of life cycle costs when using SBRSs. In addition to the acquisition costs, these include costs for maintenance and repair of the systems, costs for employees – in the investigation the helpers were permanently employed – and their training

as well as for the performance of field exercises and costs for storage and transport of the systems. The benefit of an SBRS, on the other hand, also results from the costs of damage that can be prevented during its service life, whereby a properly functioning system is assumed. An exemplary calculation of the life cycle costs of an SBRS is not carried out in the report. Only the acquisition costs of SBRSs, partly including the training of helpers (employees) by the manufacturers, for a 100 m long system with a protection height of about 1.0 m in the four categories examined — tubes, containers, freestanding barriers

and frame barriers — are mentioned.

In the frame of a Canadian study, in which the authors assessed the suitability of innovative systems as an alternative to sandbags primarily on the basis of the literature, commercial brochures, theoretical considerations and stability calculations, four different system types were examined. Among the types studied were water- or air-filled tube systems, gabion-like systems filled with sand or soil, dam beams and motorway crash barriers. Besides the assessment of the suitability of the systems, the

factors to be considered for the cost calculation of SBRS are named, but no comparative calculations are carried out. The stated costs refer to manufacturer's prices for a system with a protection length of 30 m and a protection height of about 1.0 m. The additional financial resources to be considered include costs for storage, assembly and dismantling of the systems as well as training the helpers. Moreover, the durability of the systems must also be taken into account, as a long service life has a positive effect on the number of times a system can be reused. (Biggar and Masala, 1998)

In a study conducted by the University of Kentucky (Mc Cormack et al., 2018), the possible uses of sand-filled temporary flood defence barriers to protect roads from flooding were analysed on the basis of existing operational experience. However, the systems considered are not comparable with those covered in the present study.

In Germany SBRSs have been tested according to the ANSI/ FM Approval guidelines at the TuTech Centre for Climate Impact Research – KLIFF – at the TU Hamburg on a concrete ground (Gabalda et al. 2013). The tests were mainly done on behalf of the manufactures, who have published the information only sporadically (cf. Massolle et al., 2018). Recently Popp et al. (2019) theoretically investigated the use of SBRSs to temporarily increase the height of a dyke and related costs in comparison to sandbagging. Their investigations do not relate to individual SBRSs but rather different system types (tube, basin, trestle). However, it is not clear what was included in the cost calculation. Popp et al. conclude that for temporarily raising the dyke height in case of a flood event, it is expected that SBRSs will be used more frequently due to their time-, material- and personnel saving characteristics.

None of the examples mentioned in the literature examined the functionality or costs of temporary ring dams or load drains.

SBRSs can make an essential contribution to operational flood defence owing to their functionality and time-saving characteristics as well as lower requirements for materials and personnel, and this even more so in view of the expected consequences of climate change. However, only little information is available on independent, practical tests of SBRSs, for some SBRSs no practical or independent tests are available at all, and a comparing study of the overall costs of sandbagging and SBRSs is totally missing. Both factors - functionality and economic viability - are especially relevant for decision-makers to assess the suitability of using SBRSs, which introduce great potential to make operational flood defence measures especially for disaster management much more efficient in terms of time, personnel and material. It was therefore decided to carry out systematic testing of SBRSs in the test facility of the Institute of Hydraulic Engineering at Bremen University of Applied Sciences (IWA), Germany to increase the available information on the functionality of SBRSs. The focus of the test setups was on functionality and stability as well as handling of the tested systems. First results of the test setups with regard to installation times, water heads and seepage rates have been published in Massolle et al. (2018). The present article summarises the experience gained from the test setups with regard to functionality, stability and handling of the individual systems in accordance with the guidelines for loss prevention of the German insurers for mobile flood defence systems (VdS, 2014), which are in turn based on the recommendations of the BWK (BWK, 2005), the VKF and the BWG (Egli, 2004). The system assessments obtained in this way serve to provide a practical assessment of the operational capability of SBRSs. Furthermore, the present article compares sandbagging and SBRSs in fictitious realistic scenarios in order to enable a comparison of the costs surrounding system deployment as well as the time involved and the number of helpers. The comparison serves to further clarify the practical suitability of SBRSs and, in addition to the acquisition costs, takes into account the costs respectively efforts of installing and dismantling the systems as well as logistics. In addition to the temporary flood protection dam, appropriate dyke defence measures for operational flood defence (load drain, ring dam) are also considered. The calculated operational costs always depend on the underlying system model or dimensions of the sandbag system and other factors taken

into account in the cost calculation - this necessarily calls for a certain degree of simplification. This aspect results in deviations in the findings of the above-mentioned studies by Pinkard et al. (2007) and Ogunyoye et al. (2011) and the present study.

## 2 Investigated sandbag replacement systems and equivalent sandbagging methods

The here done investigations focus on the following three operational flood protection measures: (1) temporary flood dam, (2) load drain in the case of a saturated dyke over an extensive area and (3) ring dam for reinforcement against heavy punctual exit of seepage on the inner embankment of a dyke. The classic aid for constructing these measures are sandbags. Sandbagging has proven itself during many years of application. Sandbags are made out of jute or plastic, are not standardized in size and cannot be reused once they were used in a flood event. Especially for long and/or high protection stretches a multitude of bags and sand is required. If stored properly, filled sandbags have a maximum shelf life of 5 years. If unfilled, they can be stored for up to 10 years. However, it should be noted that the shelf life of filled sandbags may be severely limited if they are stored under poor conditions. When stored outdoors, after only a few months sandbags can be so decomposed that they are no longer fit for use. Due to the longer shelf life, larger amounts of sandbags are usually stored unfilled, which in principle also minimizes the logistical efforts, because it is easier to transport large amounts of sandbags and sand separately.

In case of a flood event sandbags and/ or sand need to be transported to the scene of the flooding and if necessary, are filled manually with, e.g. shovels or with the aid of sandbag filling machines. The filled sandbags are transported to the flood defence line either with vehicles or, if the accessibility is limited, e.g. due to a poor subsoil situation, with the help of human chains, helicopters or boats. If applicable, the sandbags have to be unloaded afterwards and are put in place individually. Altogether, these steps result in a great logistical, personnel and time effort.

The construction of the three operation flood protection measures with sandbags is not standardized, slightly different techniques and quantity of sandbags might be used. However, in principal (1) the temporary flood dam is a trapezium-shaped sandbag dam in order to set up a temporary flood protection line, (2) the load drain are layers of sandbags in order to bring additional weight on the toe of a dyke and (3) is a U-shaped or circular dam in order to dam up punctual seepage at the inner embankment of a dyke.

SBRSs on the other hand require much less logistical, personnel and time effort, mainly because the systems consist of larger units, which are either not filled at all or filled with technical means, whereas the filling material water can often be gained directly at the flood defence line. Unlike sandbags, SBRSs hold the potential for a subsequent reuse during their service life. In the case of SBRSs, the guarantee period specified by the manufacturer must be compared to the actual shelf life. Inquiries to manufacturers have shown that not all producers give a guarantee or that the guarantee often only amounts to a few years. When interviewed, however, some manufacturers stated that the service life of demonstration models reached 10 years and more. Considering the materials used in the production of the SBRSs, such as tarpaulin fabric, galvanised steel or fibreglass-reinforced plastic, it can certainly be assumed that an SBRS can have a service life of 10 years and more.

Table 1 gives a short description of the investigated systems and Figure 5**Fehler! Verweisquelle konnte nicht gefunden werden.** as well as Figure 3 show the tested SBRSs in the testing facility. At least one of the container types and wall systems shown in Figure 1 was selected for each of the test setups. Flap systems could not be tested because no manufacturer was found who was prepared to provide his system for the tests. Bulk elements and panel systems were not considered. This is because in operational practice the use of bulk elements requires technical aids being available at short notice to install the elements, which is often impractical for logistical reasons or for reasons of the load-bearing capacity of the foundation soil being impaired during flooding. The use of panel systems is limited to suitable soils and low water levels. Bulk elements and panel systems were therefore not taken into account in the test setups owing to their necessity for framework conditions such as accessibility with heavy equipment and the avoidance of damage to test setups from deep ramming of retaining stakes.

**Table 1: Short description of the tested SBRSs.**

| Product Name | Description |
| --- | --- |
| **Basin system** | |
| AQUARIWA | Plates bend to cylindrical basins. If filled with water (other filling materials possible), water sacks within the basins are necessary. The bottom of the basin is covered with a plastic grid, which is welded to the plates in order to increase the stability of the basins. Spaces between the basins are sealed with a plastic tarpaulin, which is weighed down with sandbags. |
| INDUTAINER | Basin with plastic sacks, which are filled with water. The upper end is tied up. The basins are connected to each other with wooden scantlings. The space between the basins is sealed with a plastic tarpaulin, which is weighed down with sandbags. |
| Quick Damm Typ M | Open, collapsible steel frame with plastic basin filled with water. Spaces between the basins are sealed by the system itself. |
| **Trestle system** | |
| aqua defence | Hard foam panels covered with plastic tarpaulin on collapsible support elements. The tarpaulin is weighed down with sandbags. |
| Aqua Barrier | Euro-pallets covered with plastic tarpaulin on collapsible support elements. The tarpaulin is weighed down with sandbags. |
| Dam system | |
| NOAQ Boxwall | Plastic brackets, which are inserted into each other and whose connection is secured on the top side with a clamp. The underneath is sealed with a thin strip of foam on the water side and shaped in a way that any water under the system can run off to the air side. |
| **Tube system** | |
| Tiger Dam | Closed, water filled system; pyramid-shaped strapped and secured with wedges. The joints are sealed with a sleeve. Use of a plastic tarpaulin and anchoring to the ground is possible. |
| Hydrobaffle | Closed, water filled system with intermediate baffle to prevent rolling. The system is laid overlapping at the joints. |
| Mobildeich | Closed, water filled system held together as 2- to 3-tube packages with net sheathing and sealing tarpaulin. The tarpaulin is weighed down with iron chains. A geotextile is placed below the tubes. |
| Flutschutz – double – chamber tube | Closed, water filled system with two tubes of different diameter. The tubes are welded together in order to prevent rolling. A sealing mat is placed below the upstream tube. Spaces between the tubes are sealed by the system itself, whereas the joints are secured with a rope. |
| Öko-Tec Tubewall | Closed, air filled system with welded upstream skirt and plastic grid. At the underneath a drainage mat is placed between skirt and grid. The skirt is weighed down with lead belts. |

| Product Name | Description |
|---|---|
| **Load Drain** | |
| Flutschutz – load drain | Closed, water filled basin. A drainage mat is placed below the system. |
| **Temporary ring dam** | |
| Flutschutz – ring dam | Closed, water filled tube, with triangular stabilising canvas welded to the tube. |

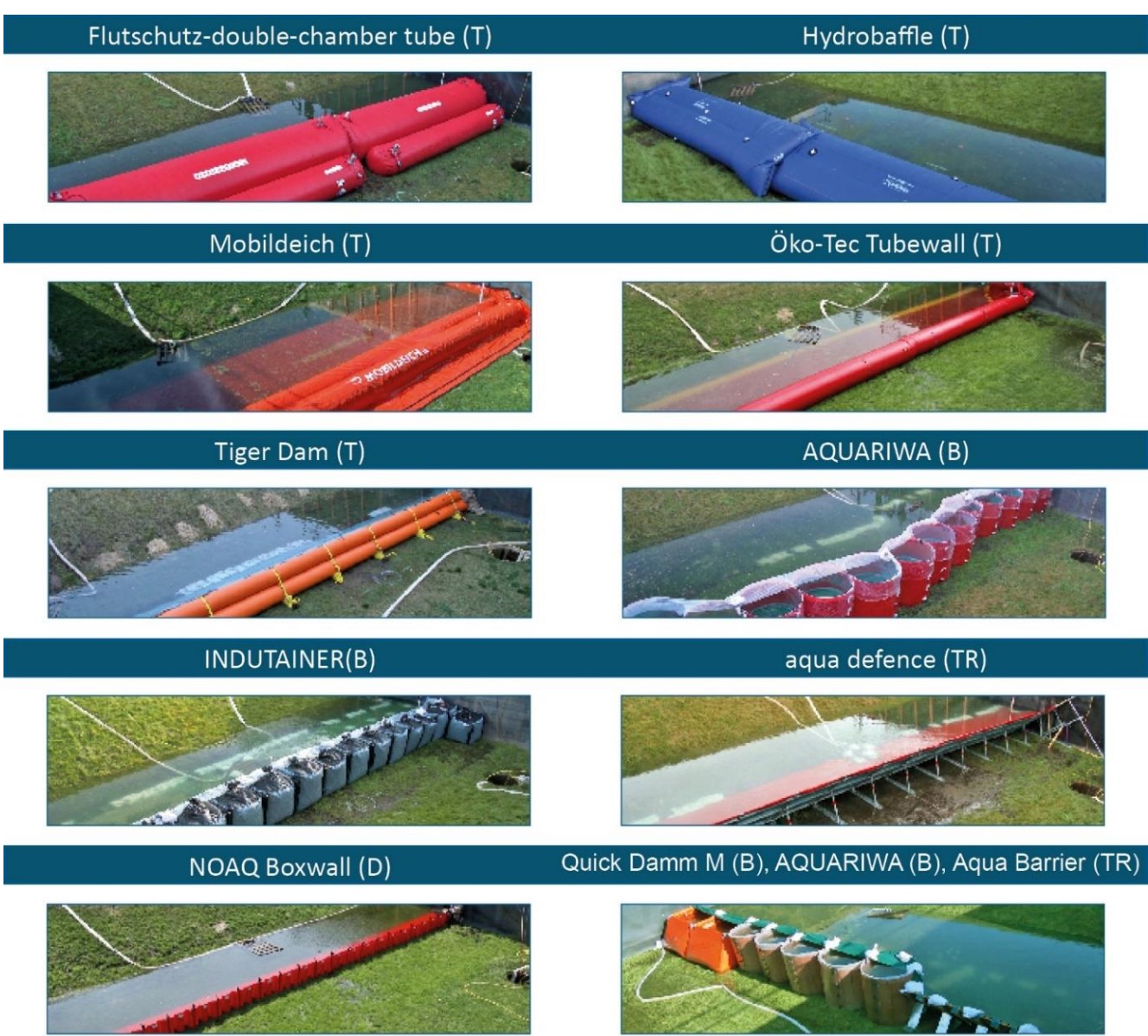

**Figure 5: The various SBRSs tested, (T) Tube system, (B) Basin system, (D) Dam system, (TR) Trestle system.**

For comparison of functionality, stability and handling, a 0.8 m high and 2.1 m wide sandbag dam was set up in the test facility

5 (Figure 6) - see Massolle et al. 2018 for further details. In addition to the linear SBRSs, Flutschutz-load drain and Flutschutz-

ring dam have been set up at the embankment of the dyke in the test facility (see Figure 3). The systems were set up on the dry and therefore stable dyke, which does not fully correspond to the reality. System dimensions and further properties of the tested SBRSs are shown in Table A1. For assembly instructions, please refer to the manufacturers' homepages.

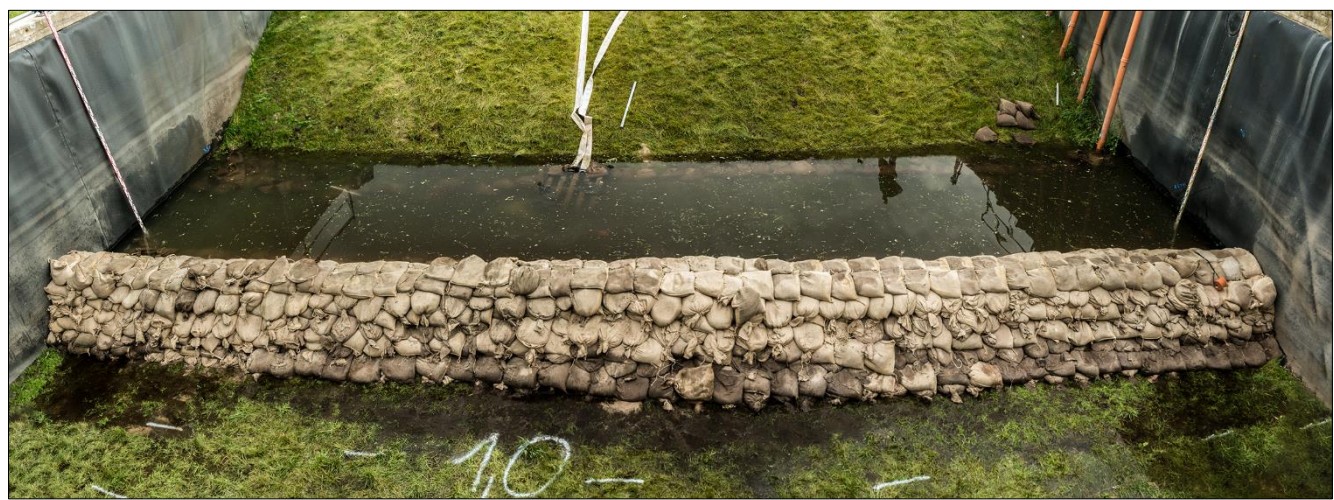

**Figure 6: Sandbag dam in the IWA test facility.**

In cases where the suppliers offered more than one system size, a variant suitable for a water head of 0.6 m was selected for the test setups. This height corresponds to the recommendations contained in the technical bulletin 'Mobile Flood Protection Systems' (BWK, 2005) for the unscheduled use of SBRSs in operational flood fighting. The recommendation results from the increasing danger of foundation-surface failure with increasing water levels. Not exceeding the specified maximum water level minimises the risk of base failure. If larger system heights are required, the risk must be weighed on a case-by-case basis. The problem is that even if a foundation expert is available at site during the flood event, restricted time and information on soil parameters do not allow an accurate analysis. Since some systems are not specifically designed for water heads of 0.6 m, over dimensioned systems such as AQUARIWA, aqua defence, Hydrobaffle and Tiger Dam were used.

The SBRSs tested are only a selection of the systems available on the market. In addition, one of the systems investigated, the Quick Damm Type M, is no longer produced, but still in use. Market analysis showed that some system types, such as basin systems and tube systems, are more frequently present on the market than others. However, the number of products of a system type does not allow conclusions to be drawn about its functionality.

Tube systems and basin systems are usually filled with water to ensure their stability. Not many tube systems or basin systems can be filled with sand. Sand fillings were not considered during the test setups as the requirements for filling and dismantling could not be met in the test facility. Therefore, only tube and basin systems filled with water were tested. The Öko-Tec Tubewall is an exception. With this system, the tube is inflated with air. The system is stabilised by a plastic sheet called 'skirt' spread out on the water side of the system, which is friction-locked to the tube. The tube is stabilised solely by the vertical hydrostatic pressure acting on the horizontally laid skirt. No other of the tested systems using a plastic tarpaulin as an upstream

skirt are connected to the system in such a friction-locking manner. A not friction-locked skirt is mostly used to improve the leak-tightness of an SBRS, which on the other hand also reduces buoyance forces under the SBRS. An upstream skirt must always be weighted down at the water-side edge, often with sandbags. The trestle and dam systems do not require filling.

## 3 Functionality, stability and handling of the tested SBRSs

### 3.1 Description of the test

The tests were carried out in the IWA test facility, which was set up on the premises of the THW Training Centre Hoya as part of the research and development project DeichSCHUTZ (2014-2017) for the development of systems to reduce buoyancy in dykes at risk of failure, funded by the German Federal Ministry of Education and Research. The facility consists of a U-shaped basin, the 15 m wide opening of which is closed by a dam (cf. Massolle et al., 2018). For the SBRS tests, the systems were set up across the entire width of the basin parallel to the dam line and the space between the dam and the system was then filled with water (Figure 7). This allows a realistic simulation of the hydrostatic load on the systems. Other possible load parameters such as current, waves, wind, flotsam and vessel impact cannot be investigated in the IWA test facility.

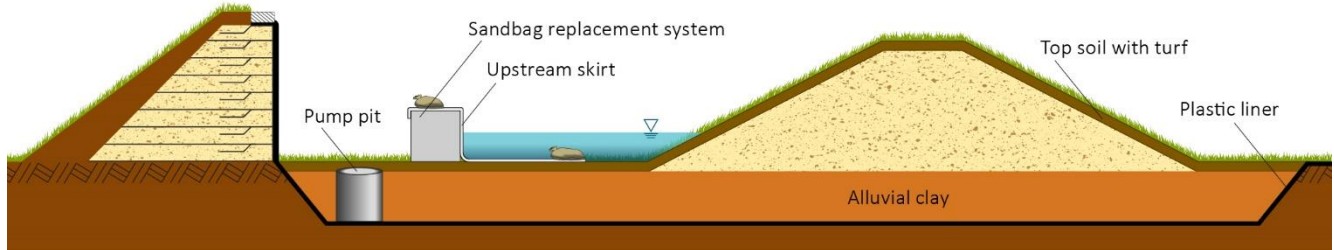

**Figure 7: Draft of the test setup in the test facility. Shown is a SBRS with upstream skirt.**

During the test setups the systems were impounded with water. Water heights were increased successive until the system failure occurred. Typical failure mechanisms of SBRSs are shown in Figure 8. The systems failed due to sliding/ rolling or tipping - stability failure did not occur in any of the systems tested. If no system failure occurred, the systems were not only completely impounded but overflowed. Occurred seepage rates - the sum of seepage through the subsoil and leakage through the system - were measured and the results have been published in Massolle et al. (2018). A SBRS should not only be functional but should also be practical in terms of handling during setup and dismantling as well as necessary space during operation and for storage, reusability and protection against vandalism just to name a few. Altogether, statements about the reliability as well as the practicability and handling of the tested systems could be derived from the test setups and related investigations.

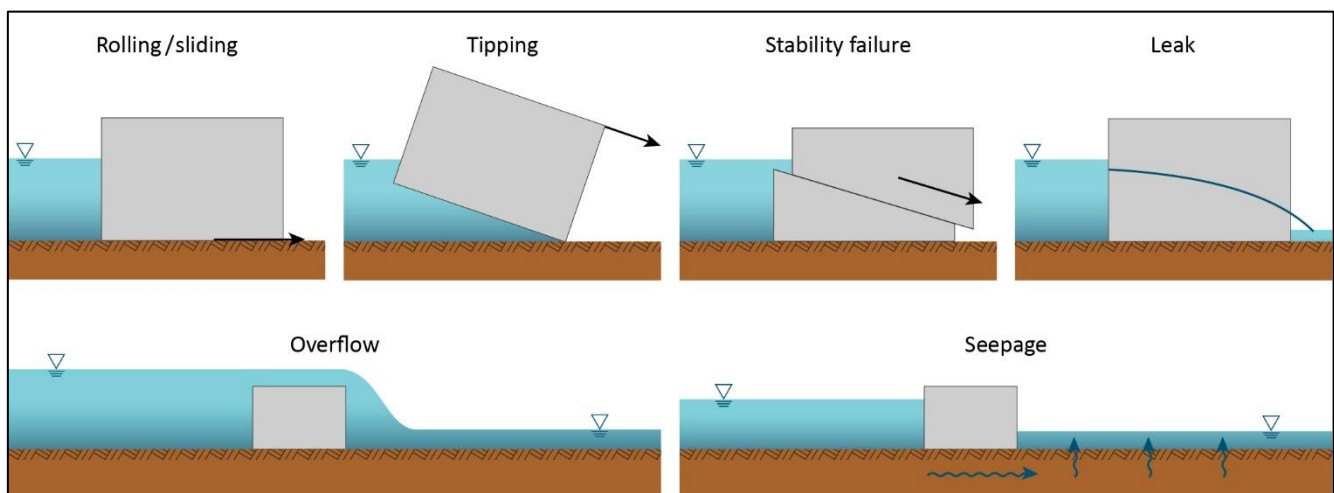

**Figure 8: Typical failure mechanisms of SBRSs (BWK, 2005, modified).**

The systems were initially dammed up to a water height of 0.6 m, in accordance with the recommendations of the BWK leaflet 'Mobile flood protection systems' (BWK, 2005). After setting a constant seepage rate at a dam height of 0.6 m (cf. Massolle et al. 2018), the water head was further increased in stages until a system failure occurred due to the water height exceeding the load limits of the system or a partial overflow of the system occurred. For an overview of the system heights and the impounded water levels achieved see Massolle et al. (2018). The Quick Damm Type M and Aqua Barrier systems were not available in sufficient length and were therefore installed in combination with the AQUARIWA system. The test basin was only briefly filled with water up to a height of 0.6 m. The NOAQ Boxwall system only has a feasible protection height of 0.5 m, but was nevertheless tested because of its simplicity and speed of installation. In principle, the manufacturer recommends the use of the NOAQ Boxwall System on paved surfaces, as this results in a better sealing effect on the underlying surface. According to the manufacturer's training material, the Tiger Dam system can be used with and without anchoring to the ground or additional plastic skirts on the water side, but is only FM-approvals certified if the skirt and the anchoring system are in place (NFBTCP, 2019). Both variants were investigated. The tightening belts pulled around the tubes were fastened in the area of every second wedge with a rope affixed by stakes on the land side and water side. Finally, a plastic skirt was spread in front of the system on the water side, which reached up to the apex of the upper tube.

Full impoundment of the tested systems and water overflow cannot be realized over the entire length of the SBRS due to unevenness of the basin floor and limited pumping capacity in the IWA test facility. This restriction is particularly relevant in

case of occurrence of an overflow load, as the unevenness meant that only a slight overflow height could be achieved in the right-hand area of the test facility (Figure 9).

If overflow occurs when using SBRSs, it must be prevented from

washing away the soil on the landside, otherwise system failure can occur. The overflowing water must be discharged or distributed over a sufficiently large area. Theoretically, an SBRS can overflow if the system is sealed via vertical water pressure, since with increasing water levels the system is increasingly held

stable via the vertical pressure. A protruding skirt on the water-side will afford more protection, as the buoyancy forces under the system are thereby minimised. Whether the system will overflow depends on its geometry and/or bulk. With increasing water levels, the probability of failure due to tilting, slipping or rolling

increases. Systems that do not benefit from the effect of vertical water pressure for stabilisation are not stabilised further with an increasing water level. In terms of stability, a high bulk and/or a

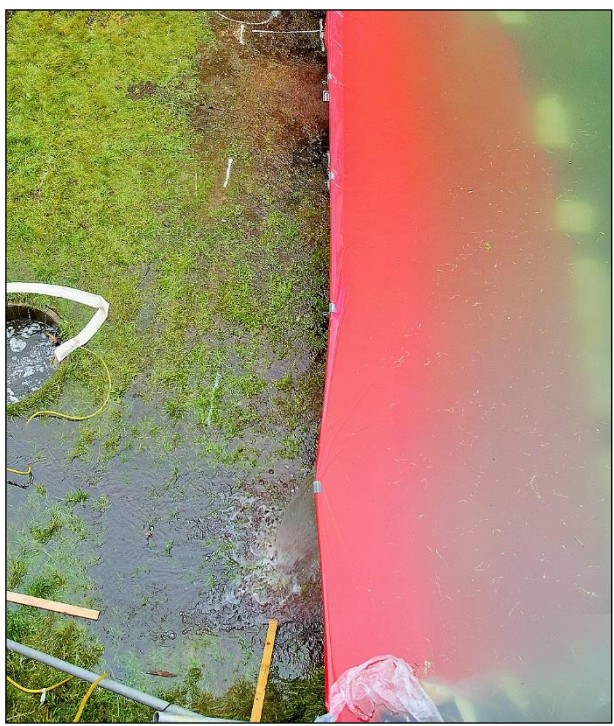

**Figure 9: Overflowing SBRS (aqua defence)**

low centre of gravity are fundamentally advantageous here. The tests do not take into account the possibility of the foundation soil giving way with increasing water levels, since damming within the test setups only took place on a defined and stable

floor. However, especially at high water levels, underground failure can be an important source of failure.

## 3.2 Test results

The systems were tested on a grass surface and were set up by two people. In some cases, there were major differences between the manufacturer's time specifications and the times measured during the test setups (cf. Massolle et al, 2018). To be set up, the systems had to be transported manually from the edge of the basin to the point of installation and thus over a maximum

distance of 15–20 m. It is quite conceivable that faster installation times can be achieved on surfaces suitable for vehicles to travel on and which offer better logistical conditions. On the other hand, significantly longer manual transport distances — and thus longer assembly times compared to the test conditions — may occur in practice. The installation times for the water-filled SBRSs also depend strongly on the available pump capacity and the water supply. In principle, however, it can be said that installation and dismantling of the systems is generally possible with just two persons and is many times faster than the

construction of a sandbag dam. In addition, it is also possible to optimise installation times by using more helpers. Systems that have no need of filling also show a clear time advantage during assembly and dismantling.

Setting up the systems is often self-explanatory and instructions are easy to follow. It is still recommended, though, to involve an expert in order to avoid possible assembly errors with their far-reaching consequences. With the Öko-Tec Tubewall system,

for example, there is a risk that the drainage mat located under the upstream skirt will be inverted, thus endangering the functionality of the system.

Taking precautions against buoyancy can be generally recommended. Systems such as NOAQ Boxwall, Tiger Dam or Öko-Tec are dependent on this safety precaution. Protection can be ensured by an upstream skirt, a drainage system, a seal on the water-side edge or anchoring of the system. Systems such as the Flutschutz-DCT-have good protection against failure owing to buoyancy as result of their high bulk weight, and no further measures are called for. However, completely weighting down an upstream skirt with sandbags or other weights is still generally recommended, as this can also considerably minimise the occurrence of seepage (cf. Massolle et al., 2018).

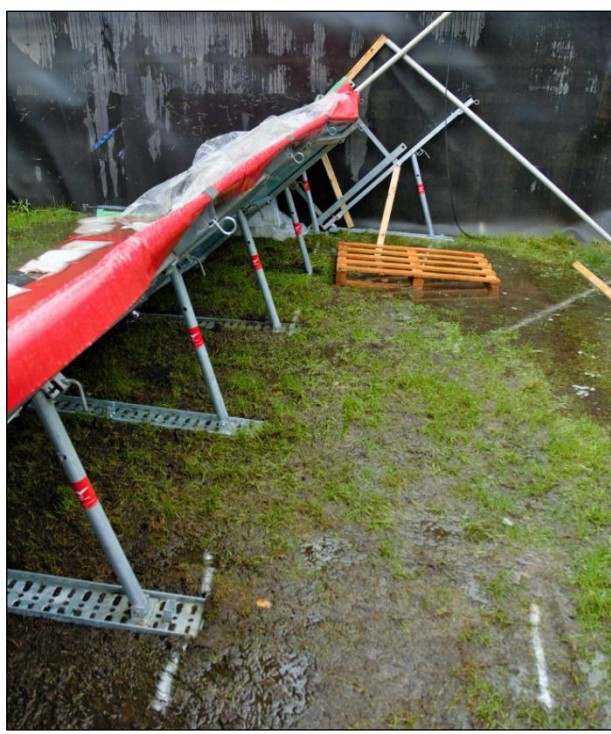

**Figure 10: Supporting columns sunk into the saturated foundation soil while damming (aqua defence).**

Especially systems with a restricted contact surface are prone to the danger of sinking into saturated ground (aqua defence, Aqua Barrier, Tiger Dam). This also applies to the AQUARIWA system, the filled base of which is flat, but whose plastic skin lies somewhat unevenly. Precise data on how long it would take for the system to fail due to sinking at the contact surfaces cannot be derived from the test carried out due to its relatively short duration of just a few hours (cf. Massolle et al, 2018). In principle, there is a correlation between the depth of subsidence, the magnitude of the load exerted, the type and the antecedent wetness of the ground underneath as well as the duration of a flood event, which can last up to several days and even weeks. Some subsidence of the systems lying on a restricted contact surface could be observed during water impoundment, but this did not lead to failure during the test setups, presumably due to the short damming time of just a few hours. Figure 10 shows the aqua defence system during dismantling. The system sank the deepest into the foundation soil in the area of the greatest water depths during damming - at the top of the picture. In this area, however, the system also overflowed while the test basin was being filled with water, so that some of the increased subsidence was probably due to erosion of the foundation soil.

Particularly in the case of fine sandy soils, there is a risk of foundation soil failure due to hydraulic heave or erosion caused by water flowing under the system. Especially when additional pumping is used, care must be taken that the soil under the systems is not removed with the flow of water being pumped out. There is also a risk that the friction between soil and system on paved ground will be reduced by the presence of loose grains of sand or gravel. Here, it is recommended to sweep the areas around the contact surfaces prior to installation. Minor unevenness can be levelled out with sandbags or lime that swells in contact with water. When installing the systems, attention must be paid to whether there are gradients in the terrain across or along the

planned system line, as this would increase the risk of tipping, sliding or rolling. Some systems shifted or were deformed when the test basin was being filled with water, owing to play in their construction or expansion of the material they are made of, but then stabilised again (Flutschutz-DCT, Hydrobaffle, Tiger Dam, Aqua Barrier). The pending failure of all the tested systems when overloaded was always indicated by visible shifting, but this was usually so quick that there was no possibility of taking countermeasures over longer lengths.

In terms of seepage rates, the tested systems are either comparable to a sandbag dam or to a sandbag dam with protruding plastic skirt (cf. Massolle et al. 2018).

In summary, it can be stated that all the systems tested remained stable at the water levels specified by their manufacturers (Figure 11). The systems aqua defence, NOAQ Boxwall, Mobildeich, Öko-Tec Tubewall (Öko-Tec TW) as well as Tiger Dam with anchoring and skirt (Tiger Dam with A.) held a full water head with low incidence of overflow. The systems we could not dam up to maximum capacity (AQUARIWA, INDUTAINER, Flutschutz-DCT, Hydrobaffle) were capable of reaching higher water levels than those specified by the manufacturers. The Tiger Dam tube system was only able to achieve the protection height of 0.6 m specified by the manufacturer by the additional use of an upstream skirt and anchoring to the ground: a test setup without skirt and anchoring threatened an early system failure. The Quick Damm Type M and Aqua Barrier systems were not available in sufficient quantities and could only be tested in combination with the AQUARIWA system. Therefore, water was only dammed up to a height of 0.6 m. Since the tests were carried out without any further loads caused by currents, waves, flotsam, etc., the possibility of increasing the protection heights given by the manufacturers cannot be deduced. Table 2 summarises the advantages and disadvantages of the various system types determined in the frame of our test setups.

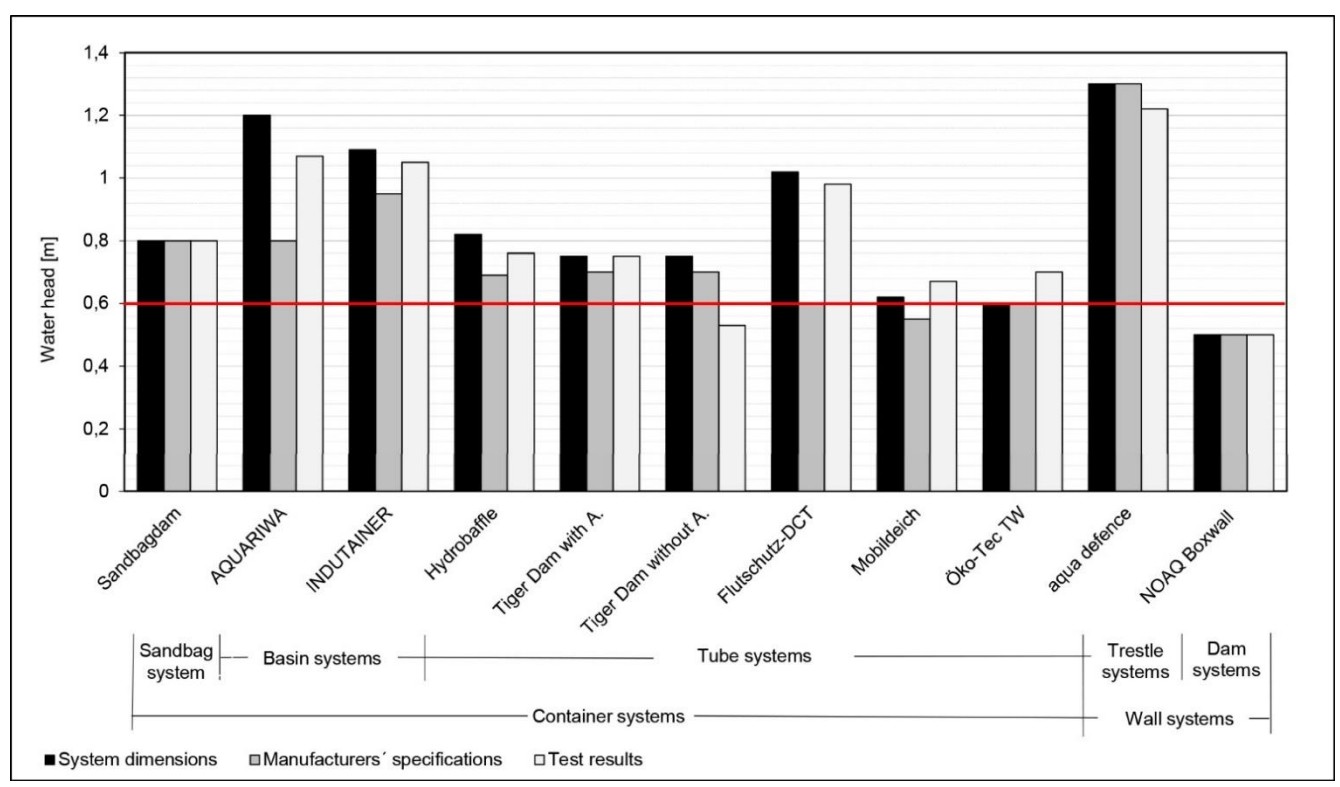

**Figure 11: Water levels achieved during the test setups (Massolle et al., 2018). The red line marks the maximum water height of 0.6 m, which is recommended for the unscheduled use of SBRSs (BWK, 2005).**

**Table 2: Summary of the most important advantages and disadvantages of different system types.**

| Basin system | | |
|---|---|---|
| Advantage | - | High stability even no or small volumes of retained water (with influence of wind or similar) |
| | - | Seals well even with low volume of retained water |
| | - | Offer high safety with sand filling |
| Disadvantage | - | Installation time |
| | - | Filling material |
| **Tube system** | | |
| Advantage | - | High stability even no or small volumes of retained water (with influence of wind or similar) |
| | - | Seals well even with low volume of retained water |
| Disadvantage | - | Installation time |
| | - | Filling material |
| **Flap, trestle, dam systems** | | |
| Advantage | - | Installation time |
| | - | No filling material |
| | - | Usually overflowable |
| Disadvantage | - | Good stability only with increasing height of retained water (problematic with wind influence or similar) |
| | - | Good seal only with higher levels of retained water |

The system dismantling of the tested SBRS was generally uncomplicated. In the case of water-filled systems, it must be ensured that the number, position and size of the openings for emptying the systems significantly influence the emptying time as well as the possibility of simple complete emptying. Even if only a small amount of residual water remains in the system, the resulting weight can exceed a manageable level. All systems must always be cleaned and dried before being stored for reuse.

The INDUTAINER system may be considered as a disposable system, as cleaning or drying is difficult owing to its intricate design. However, it has a comparatively low purchase price, so that the use of the system can be economical even if only used once. Some other SBRS also have limited disposal costs after use. This applies in particular to systems in which the upstream skirt is (preferably) to be weighted down with sandbags. The required sandbag requirement, however, is low at approx. four sandbags per metre.

These tests, though, were carried out under idealised conditions using a bundle of wooden slats as flotsam. Since the failure of an SBRS threatens the flooding of the hinterland with a correspondingly high damage potential and SBRSs are to be regarded as more susceptible to mechanical impacts and vandalism due to their design, these aspects should be evaluated particularly critically. Mechanical effects and vandalism, though, are also relevant when using sandbag systems. In the opinion of the authors, these aspects should therefore not be an exclusion criterion, despite their particular relevance for SBRSs. However, it

is advisable to make higher demands on monitoring of the systems during their use.

The Guidelines for Loss Prevention issued by German Insurers for Mobile Flood Protection Systems (VdS, 2014) contain a specimen evaluation form for SBRSs, which is intended to serve as a decision-making aid for system evaluation for persons responsible for flood defence. The SBRSs tested were evaluated in accordance with these guidelines, for comparison the sandbag dam was also evaluated according to these guidelines (Table 3). For sandbagging, where applicable, the evaluation is

comparable for sandbag dam, load drain and temporary-ring dam. The evaluation criteria relate to the area of application, stability, procurement and durability, installation, dismantling and maintenance as well as the logistics surrounding the systems. If a specification could not be determined or derived from the results of the test setups, manufacturers' specifications were used, or the evaluation was carried out on the basis of theoretical assessments. The failure mechanisms affecting the surface an SBRS is installed on, such as caused by hydraulic heave or erosion, were not considered due to their dependence

on the variable site conditions encountered in operational practice. Also not taken into consideration were the system connections to walls or the like, the possibility of laying the system in curves or with angles or the system behaviour on different substrates (soft, solid, rough, smooth, even, uneven, permeable, impermeable etc.). The criteria on which the system evaluations are based are described in Table 4.

**Table 3**: System Evaluation; DCT: Double-chamber tubeTW: Tubewall, TD: Tiger Dam, A: Skirt and Anchoring, RD: Ring dam, LD: Load drain.

| | Sandbagging° | AQUARIWA | INDUTAINER | Quick Damm | Aqua Barrier | aqua defence | NOAQ Boxwall | Flutschutz-DCT | Hydrobaffle | Mobildeich | Öko-Tec TW | TD without A. | TD with A. | Flutschutz-RD | Flutschutz-LD | Explanation / Remarks |
|---|---|---|---|---|---|---|---|---|---|---|---|---|---|---|---|---|
| **Application area** | | | | | | | | | | | | | | | | |
| Uneven ground | + | - | - | o | o | o | o | o | + | + | o | + | + | + | + | Test/ own estimate |
| Unsurfaced ground | + | - | - | + | - | - | o | + | + | + | + | o | o | + | + | Test/ own estimate |
| Height of retainable water (h) | + | o* | o | o | o | o | - | o | +* | +* | o* | -* | +* | / | / | Test/ * Manufacturer's data, e.g. not all system heights tested |
| Height adjustable | + | - | - | - | - | - | - | - | - | o | - | o | o | / | o | Manufacturers' data |
| Overflowable | n/s | o* | - | n/s | o | o | + | - | - | + | + | - | + | + | / | Test/ * Perchance with sand filling |
| Installation in water | + | o | - | - | o | o | o | - | +* | +* | - | - | - | / | / | Own estimate/ * Manufacturer's data |
| Space requirement in use | o | - | - | o | - | - | + | - | o | - | - | + | - | / | / | Manufacturer's data |
| **Stability** | | | | | | | | | | | | | | | | |
| Tipping stability | + | - | - | o | o | o | o | + | + | + | + | o | + | / | / | Test & own estimate |
| Roll / slide stability | + | + | o | o | + | + | o | o | - | + | o | - | + | o | o | Test & own estimate |
| Buoyancy stability | + | + | + | o | + | + | o | o | - | + | o | - | + | / | / | Test & own estimate |
| Anchoring | - | - | - | - | o | o | - | - | - | - | + | / | + | / | / | Manufacturer's data |
| Resistance against mechanical effects | + | o | o | o | o | o | - | o | - | + | - | - | o | +* | +* | Own estimate/ *if, only from landside |
| Resistance against vandalism | - | - | + | - | - | - | - | - | - | - | - | - | - | - | - | Own estimate |
| Domino effect | + | + | - | + | o | o | - | - | o | - | - | - | - | / | / | Own estimate |
| **Procurement and durability** | | | | | | | | | | | | | | | | |
| Costs | + | o | + | n/s | o | - | o | o | o | o | - | + | + | - | - | Manufacturer's data |
| Service life | o | o/+*** | -* | n/s | n/s | n/s | o** | + | + | o** | o** | + | + | + | + | * During continuous operation/ own estimate  ** Legal warranty/ Manufacturer's data  *** o: Water sack/ Manufacturer's data  +: GRP panel/ Manufacturer's data |
| Reusability | - | o | o | + | + | + | + | + | + | + | + | + | + | + | + | Manufacturer's data |
| **Installation** | | | | | | | | | | | | | | | | |
| Installation time | - | o* | o* | n/s | + | + | + | o* | o* | o* | + | -* | -* | +* | +* | Test/ * According to pumping capacity |
| Equipment requirement | - | - | - | o | o | o | + | o | o | o | o | o | - | o | o | Manufacturer's data |
| Persons | - | + | + | + | + | + | + | + | + | + | + | + | + | + | + | Manufacturer's data or own estimate |
| Requirement of filling material | - | o* | o | o* | + | + | + | - | o | o | o | o | o | o | o | Manufacturer's data/ * with sand filling |
| Number of individual elements | - | - | o | + | + | o | + | + | + | o | o | - | - | + | + | Manufacturer's data |
| Simplicity of installation | + | + | + | + | + | o | + | + | + | + | o | - | - | + | + | Tests |
| Weight of individual elements | + | + | + | o | + | + | + | o-* | o-* | o** | o | o | o | + | o | Man.'s data/ * according to system length **with reel |
| **Dismantling and maintenance** | | | | | | | | | | | | | | | | |
| Simplicity of dismantling | + | o* | + | o* | + | + | + | o | + | + | + | + | + | o | + | Test/ * sand filling - own estimate |
| Disposal effort | - | o* | o | o* | - | o | + | + | + | + | + | o | + | + | + | Manufacture's data/ * sand filling |
| Cleaning effort | / | o | - | o | o | o | o | o | o | o | o | o | o | o | o | Own estimate |
| Repairs and spares | / | + | - | + | + | + | - | + | + | + | + | + | + | + | + | Own estimate |
| **Logistics** | | | | | | | | | | | | | | | | |
| Space for storage/ transport | - | + | + | o | + | + | + | o | o | o | o | + | + | o | o | Manufacturer's data |

**Legend**

+ = good          o = medium          - = bad          / = not relevant          n/s = not specified

° For sandbagging the manufacturer's data is based on own estimates

**Table 4**: Evaluation criteria.

| | Evaluation criteria |
|---|---|
| **Area of application** | |
| Uneven ground | Applicable on unevenness, curbstones, etc. |
| Unsurfaced ground | Special requirements for the condition of the foundation surface |
| Height of retainable water | Height of retainable water h up to 0.6 m = -; up to 1.5 m = o; up to 3.0 m = + <br> Observe recommendations for unscheduled use of SBRS according to BWK (2005) |
| Height adjustable | Subsequent increase possible |
| Overflowable | Overflow capability according to manufacturer (M) or determination in authors' tests (AT) <br> No = -; Yes (AT or M) = o; Yes (AT and M) = + |
| Installation in water | Manufacturer's specification or own estimate based on system characteristics |
| Space requirement in use | Depth incl. any upstream skirt ≤1,0 m = +; ≤2,0 m = o; >2,0 m = - (refers to the system variants tested) |
| **Stability** | |
| Tipping stability | Tube systems are less prone to tipping than dam or trestle systems. The heavier the installed systems, the less prone they are to tipping. (Selective) Sinking into the ground increases the risk of tipping. Anchoring or securing against buoyancy counteracts tipping. |
| Roll / slide stability | Tube systems are generally more susceptible to rolling away. The lower the weight and the smoother the foundation surface of the system, the easier it is for the system to slip. Anchoring or securing against buoyancy counteracts sliding or rolling. Flutschutz load drain and ring dyke always have to be positioned partly on the horizontally plane in front of the landside dyke embankment. |
| Buoyancy stability | The risk of system failure due to buoyancy is greater for filled systems with a lower weight. Depending on the shape, buoyancy forces can also act on the water side (e.g. tube systems). Systems with a large foundation surface which achieve their load bearing effect through the vertical water pressure from the outside also have a greater risk of failure due to buoyancy. An upstream skirt, drainage, seal or anchoring counteracts failure caused by buoyancy. |
| Anchoring | System can be anchored against wind, current, slipping or rolling |
| Resistance to mechanical effects | Susceptibility to damage e.g. by flotsam impact |
| Resistance against vandalism. | Susceptibility to deliberate damage |
| Domino effect | Threat to the entire dam due to failure of individual elements |
| **Procurement and durability** | |
| Costs | ≤100 €/m = +; ≤300 €/m = o; >300 €/m = - (refers to the system variants tested) |
| Service life | Service life according to manufacturer ≤1 year = -; ≤5 years = o; >5 years = + |
| Reusability | Manufacturer's data |
| **Installation** | |
| Installation time | Installation time according to manufacturer or from own test. For all water-filled systems, the installation time depends on the pump used. |
| Equipment requirement | Tarpaulins, sandbags, hoses, pumps, adapters or blowers <br> Tarpaulin and etc. = -; Tarpaulin or etc. = o; no equipment requirement = + |
| Persons | ≤2 Persons = + |
| Requirement of filling material | Sand filling = - ; water filling = o; no filling = + |
| Number of individual elements | Number of individual parts |
| Simplicity of installation | System installation easy to understand and to perform |
| Weight of individual elements | ≤35 kg = +; ≤100 kg = o; >100 kg = - (refers to the tested system variants) |
| **Dismantling and maintenance** | |
| Simplicity of dismantling | System dismantling easy to understand and easy to perform |
| Disposal costs | Foils, tarpaulins, sandbags - Disposal after use |
| Cleaning costs | Effort involved in system cleaning |
| Repairs and spares | Minor damage can be repaired by the user. Material and spare parts are available. |
| **Logistics** | |
| Space for storage/ transport | Compactness of the dismantled system |

## 4 Costs of deployment, time involved, helpers and logistics

### 4.1 Description of scenarios

The costs as well as time, helper and logistic requirements for the installation and dismantling of sandbag systems and SBRSs were determined for the following three different cases:

1. Temporary flood dam
2. Load drain in the case of a saturated dyke over an extensive area
3. Ring dam for reinforcement against heavy punctual exit of seepage on the inner embankment of a dyke

In case 1, in addition to the sandbag dam, three different SBRS types (basin, tube and trestle) were considered. Regarding the temporary flood dam, based on the experiences of the test performances described in section 2, one manufacturer of each system type was selected. However, there was more than one suitable system of each system type but the scope of the investigations had to be limited due to financial and temporal reasons. Regarding their function, protection against flooding, based on the experience of the test setups the chosen systems can be seen as equivalent to sandbagging. Although the systems show different safety margins, but the degree of safety can only be defined in detail knowing relevant parameters such as the coefficient of friction, which have been outside the scope of the analysis carried out. In cases 2 and 3, the only suitable SBRSs on the market are provided by Flutschutz. The system performances on the dry dyke were in accordance with the manufacturer's statements. Furthermore, the mode of action of the corresponding SBRSs is the same as for sandbagging. The authors therefore assume the SBRS Flutschutz-load drain and Flutschutz-ring dam equivalent to sandbagging, not taking into account possible differences in safety margins. When determining the costs for the installation and dismantling of the systems, in addition to the acquisition costs, the costs for logistics (hiring the truck, fuel, driver, repair) and helpers were taken into account as well as the costs of materials (sand, sandbags respectively acquisition cost for SBRSs, including component parts) and the disposal of sand and sandbags.

In the case of the temporary flood dam, a protection length of 100 m and a protection height of 1.0 m were assumed. The height of the sandbag dam was assumed to be 1.0 m, as the dam can theoretically protect against water levels up to its full height. The SBRS AQUARIWA (basin system) with a protection height of 1.0 m and a freeboard of 0.5 m, the Flutschutz-DCT with a protection height of 0.6 m and a freeboard of 0.3 m as well as aqua defence (trestle system) with a maximum protection height of 1.3 m (identical to system height) were compared. The differences in the protection heights are system specific and cannot therefore be avoided. The practical tests (cf. Massolle et al., 2018) have shown that the Flutschutz-DCT can dam a water head up to a height of 1.0 m, whereby, due to the lateral pressure exerted when filling the test basin, performance can be increased above the system height of 0.9 m specified by the manufacturer. In case 2, one Flutschutz- load drain was compared with the equivalent length of a sandbag load drain, and in case 3, one Flutschutz- ring dam was compared with one sandbag ring dam (see Figure 3).

All cost calculation assumed technical assistance provided by the disaster services of the German Federal Agency for Technical Relief (THW). Such federal assistance takes place within the framework of inter-agency cooperation and is generally requested

by the responsible state authorities during extreme flood events in Germany. For the resources made available — primarily vehicles, pumps and hoses — as well as THW helpers, the costs were calculated on the basis of the Ordinance on the Implementation and Invoicing of Assistance provided by the THW (*Verordnung ueber die Durchfuehrung und Abrechnung von Hilfeleistungen des Technischen Hilfswerks*, in accordance with the Annex to Section 4 (3) of the THW Invoicing Ordinance (THW-V, 2019). During a flood, the German Federal Armed Forces and other relief organisations such as fire brigades and the police can be deployed in addition to THW. Depending on the organisation, the individual costs may vary: this, however, has not been taken into consideration for the present cost estimate.

The distance between the filling station for sandbagging respectively the place of storage of SBRSs and the site of operation is 5 km, i.e. 10 km for one round trip. Optimum access to the site of operation allows the use of trucks. Due to the heavily soaked subsoil in case 2 and 3, the access from the dyke defence road to the dyke toe is limited, therefore additional helpers to form a sandbag chain and pass on the sandbags to the dyke are needed. The comparable SBRSs in case 2 and 3 can be carried to the dyke by two persons. The operation is carried out with THW personnel and means, i.e. trucks, as well as pumps and hoses for the water filled SBRSs are provided by the THW. Furthermore, it is assumed that the travel distances for installation and dismantling of the systems are the same length. That is why the logistics of installation and dismantling show no differences.

The requirement for sandbags and sand as well as the labour needed for filling and laying the sandbags are based on empirical values supplied by THW (THW, 2017). The labour time needed for the installation of the SBRS was estimated on the basis of the authors' empirical values (cf. Massolle et al, 2018). In the case of water-filled systems in particular, the time required to dismantle an SBRS is less than that required for the installation, as the systems can be allowed to drain empty at the same time without the need for pumps. For the water-filled systems, 20% of the time required for installation was therefore estimated for dismantling. In practice, it should be noted that these estimates depend on the conditions and accessibility on site and, moreover, at least in Germany that dismantling is generally not financed by the federal authorities and therefore also not by THW. With the end of the flood hazard — and thus the disaster event — assistance on the part of the federal authorities is terminated: the municipalities and administrative districts become responsible for the measures taken. Owing to a lack of helpers, this can often lead to considerable problems following major flood events.

The following times were assumed for cleaning the systems:

- Flutschutz-DCT, length 10 m: 1.5 h
- aqua defence, length 1.22 m: 5 min
- AQUARIWA, length 1.5 m: 5 min
- Flutschutz-load drain: 1 h
- Flutschutz-ring dam: 1 h

The sandbag requirement for SBRSs with upstream skirt (AQUARIWA, aqua defence) is 4 sandbags per linear metre. The basic helper requirement is 10 persons for sandbagging and 2.5 persons per SBRS, whereby foremen (group leaders, i.e. lower command) are taken into account. In case of SBRSs group leaders can take care of two different areas of application – therefore

only half a helper is counted for the lower command per 100 m for the installation of a SBRS. The other two helpers are installing the SBRS, resulting in 2.5 persons per SBRS. In practice, the systems should be set up by a larger team of helpers, but fictitious helper teams with a minimum number of helpers were assumed for the calculation. Per helper hour, 22.00 € is estimated as the average loss of remuneration to be reimbursed (THW-V, 2019). The average weight of a sandbag is 12 kg

(THW, 2017). A requirement of 15 kg sand per sandbag was assumed in order to take overfilling and sand losses into account. On the other hand, no reserve margin for defective sandbags etc. is taken into account, but is considered to be included in the excess demand for sand. A sandbag purchase price of 0.20 € takes into account the slight price increase to be expected during a flood event, sand is calculated with a price of 10.5 €/t. Travel costs were assumed to be 1.52 €/L diesel and 25 L/100 km. No voluntary or private-sector assistance is taken into account. However, the participation of other volunteers, for example local

people, can significantly reduce the costs for the construction of a sandbag dam, as the helper costs make up the largest cost factor. It should be taken into account, though, that in case of volunteers from the local population, the resulting costs are usually borne by the volunteers themselves — the costs are therefore only transferred. The calculation also does not include costs for travel/ food/accommodation/ sanitary needs of the helpers, upper command, long transport routes/ alternative means of transport in case of poor access, other material requirements (shovels etc. for filling the sandbags), the transport of sand/

supplementary materials as well as storage of SBRSs/ sandbags/ shovels etc. and necessary repairs to SBRSs.

In principle, the selected SBRSs are reusable. Only the AQUARIWA system needs to have the inner bags replaced after using the system; the price per bag is low and was therefore neglected in the calculation. However, to replace worn off elements 5% of the investment costs are estimated. It is assumed that with smaller quantities of SBRSs, storage on site, e.g. by local dyke management units (*Deichverbaende*), is possible without difficulty. Only in the case of larger stocks higher demands are placed

on storage capacities. Just like SBRSs, sandbags must be stored but they have a significantly lower shelf life than SBRSs (see chapter 2). In view of this, the calculation equates the repair requirements of SBRSs with the inspection and renewal requirements of stored sandbags.

The need to regularly test the construction of SBRSs is likewise equated with the requirement to carry out flood protection exercises when relying on the use of sandbag systems. It was also assumed that the sandbag systems, like the SBRSs, should

be continuously monitored during a flood event in order to monitor their functionality and to check the systems for damage caused by mechanical influences or vandalism. If deemed appropriate, the SBRSs should be inspected at shorter intervals than sandbag systems. However, the additional requirement for labour is comparatively low and was therefore neglected.

### 4.2 Costs of deployment

The overview of the total cost of installing and dismantling the flood protection systems shows that under the assumed

conditions the costs resulting from the one-off use of the SBRSs are around 30 %-50 % higher than for sandbagging. However, since the SBRSs, in contrast to sandbags, are largely reusable, the higher investment costs of the SBRSs are already amortized during their second application. Table 5 shows the cost estimates for the temporary flood dams (case 1) and Table 6 for the load drain (case 2) and the ring dam (case 3). In each case, the costs incurred for installing the systems exceed the costs for

their dismantling. Whereas the costs for dismantling the sandbag dam amount to approx. 70 % of the costs of installation, in the case of SBRSs the dismantling costs are in the low single-digit percentage range compared to their installation.

**Table 5: Comparison of the costs for installation and dismantling of sandbagging and SBRSs – temporary flood dam, protection length 100 m (case 1).**

| | Sandbag dam | Flutschutz -DCT | aqua defence | AQUARIWA |
|---|---|---|---|---|
| Helpers, incl. lower command | 10 | 2.5 | 2.5 | 2.5 |
| Sandbag requirement [40 x 60 cm, empty] | 16 500 | - | 400 | 400 |
| **Installation** | | | | |
| Time per dam [h] | 61.88 | 7.50 | 8.48 | 10.71 |
| Costs of helpers [€] | 13 612.50 | 412.50 | 466.40 | 523.05 |
| Costs of materials, incl. replacements [€] | 5 898.75 | 42 930.33 | 47 400.15 | 51 758.87 |
| Costs of trucks, incl. fuel [€] | 641.47 | 35.06 | 37.56 | 28.02 |
| Total installation costs without materials [€] | 14 253.97 | 447.56 | 503.96 | 617.07 |
| 3% sundry costs [€], based on total operating costs: 15 € - 150 € | 150.00 | 15.00 | 15.12 | 18.51 |
| Total costs of installation [€] | 20 302.72 | 43 392.89 | 47 919.23 | 52 416.95 |
| **Dismantling** | | | | |
| Time per dam [h] | 20.63 | 16.55 | 12.96 | 9.10 |
| Costs of helpers [€] | 4 537.50 | 907.50 | 712.8 | 390.61 |
| Costs of materials [€] | 8 250.00 | - | 200.00 | 200.00 |
| Costs of trucks, incl. fuel [€] | 641.47 | 35.06 | 37.56 | 28.02 |
| Total dismantling costs without materials [€] | 5 178.97 | 942.56 | 750.36 | 418.63 |
| 3% sundry costs [€] based on total operating costs: 15 € - 150 € | 150.00 | 28.28 | 22.51 | 15.00 |
| Total costs of dismantling [€] | 13 578.97 | 970.83 | 972.87 | 633.63 |
| **Installation and dismantling** | | | | |
| Total costs [€] | 33 881.69 | 44 363.72 | 48 892.10 | 53 050.58 |

**Table 6: Comparison of the costs for the installation and dismantling of sandbag and sandbag replacement systems load drain (case 2) and ring dam (case 3).**

| | Load drain | | Ring dam | |
|---|---|---|---|---|
| | Sandbag | Flutschutz | Sandbag | Flutschutz |
| Helpers, incl. lower command | 10 | 2.5 | 10 | 2.5 |
| Sandbag requirement [40 x 60 cm, empty] | 980 | - | 900 | - |
| **Installation** | | | | |
| Time per element [h] | 4.90 | 0.50 | 4.50 | 0.50 |
| Costs of helpers [€] | 1 078.00 | 27.50 | 990.00 | 27.50 |
| Costs of materials, incl. replacements [€] | 350.53 | 3 046.28 | 321.75 | 3 726.01 |
| Costs of trucks, incl. fuel [€] | 41.31 | 6.93 | 38.18 | 6.93 |
| Total costs without materials [€] | 1 119.31 | 34.43 | 1.028.18 | 34.34 |
| 3% sundry costs [€] based on total operating costs: 15 € - 150 € | 33.58 | 15.00 | 30.85 | 15.00 |
| Total costs of installation [€] | 1 503.24 | 3 118.21 | 1 380.78 | 3 748.51 |
| **Dismantling** | | | | |
| Time per dam [h] | 2.45 | 1.10 | 2.25 | 1.10 |
| Costs of helpers [€] | 539.00 | 60.50 | 495.00 | 60.50 |
| Costs of materials [€] | 490.00 | - | 450.00 | - |
| Costs of trucks, incl. fuel [€] | 41.31 | 6.93 | 38.18 | 6.93 |
| Total operating costs without materials [€] | 580.31 | 67.43 | 533.18 | 67.43 |
| 3% sundry costs [€]based on total operating costs: 15 € - 150 € | 17.41 | 15.00 | 16.00 | 15.00 |
| Total costs of dismantling [€] | 1 087.72 | 82.43 | 999.18 | 82.43 |
| **Installation and dismantling** | | | | |
| Total costs [€] | 2 590.96 | 3 200.63 | 2 379.96 | 3 880.36 |

In the case of sandbagging, both sand and sandbags must first be procured. These are usually only stocked in limited quantities, and in the event of procurement during a flood event, it must be expected that prices will rise sharply, so that they can even exceed the here assumed cost of sandbags. The sandbags must then be filled and laid with a great deal of time and effort. These aspects must be weighed against the investment costs for the respective SBRS, which, however, can be used several times. In order to replace damaged systems after use, an average new procurement requirement of 5% is assumed within the system service life. The sandbags required to weigh down and seal the upstream skirt of an SBRS are comparatively insignificant. The logistics costs for installation and dismantling are quite the same owing to the equally long travel distance: for sandbagging they are higher compared to SBRSs, owing to the greater bulk. Basically, the logistics costs for all systems are comparatively low, which is also due to the comparatively low costs for the here assumed use of THW vehicles. When dismantling, the costs for sandbagging are higher than for the SBRSs, owing to the extra need for helpers and the disposal of sandbags. However, if it is possible to deploy heavy equipment for the dismantling of a sandbag dam, these costs can be lower than estimated in the present calculation because of the lower requirement for helpers and the shorter time involved. Overall, the largest cost items

for sandbagging are the costs for the deployment of helpers and the procurement of materials (sand, sandbags), and for the SBRSs the procurement of the systems. If, in addition to the costs for installation, the costs for dismantling are also taken into account, from a financial point of view and under the assumed conditions, the purchase of SBRSs makes sense as they are amortised already during the second deployment. The investment costs did not include a quantity discount for the purchase of

larger system lengths.

From a financial point of view, the use of SBRSs as a temporary flood dam is particularly worthwhile for protection against higher flood levels. If the protective height is reduced, the installation costs for the temporary sandbag dam decrease owing to the lower sandbag requirement. SBRSs, on the other hand, can rarely be flexibly adjusted in height, so that with lower system heights, the cost amortization in comparison to sandbag dams of low height only takes place after a number of deployments.

For example, the costs for constructing a sandbag dam with a height of 0.50 m and a length of 100 m are only approx. 8 090 € for installation, approx. 5 352 € for dismantling and approx. 13 442 € for installation and dismantling. If an SBRS is offered in different system heights, savings can also be expected if lower system heights are used, but these are less significant. It should also be noted that the procurement costs of SBRSs supplied by other manufacturers may differ from those of the manufacturers considered here.

If there should be insufficient water available from natural sources (e.g. river water) in the immediate vicinity of where water-filled systems are to be installed, the costs for the water filling of hydrants are comparatively low (approx. 400 € Flutschutz-DCT and 150 € AQUARIWA). If tank trucks have to be used, however, the logistical effort increases. Notwithstanding, the time, material and helper advantages of SBRSs remain in all of the cases considered here.

The calculations did not take into account the costs for upper command or travel, meals, overnight accommodation and sanitary

requirements of the helpers. For upper command, i.e. the disaster control management, technical incident command and platoon, 5 € per helper in the lower command and day can be assumed. The costs for upper command are realistic overhead costs related to the number of helpers in action. With an estimate of 25 € per day for overnight accommodation, food and sanitary needs of the helpers, then with a helper day of 12 hours per sandbag system in cases 1, 2 and 3 approx. 6 %, and per SBRS approx. < 1 % more costs are incurred.

**4.3 Time, helper and logistics requirements**

For cases 1, 2 and 3, the estimated time, helper and logistics requirements are shown in Table 7 and Table 8. Time 'materials' refers to the time needed to fill the sandbags – aqua defence and AQUARIWA need sandbags in order to weight down the upstream skirt. Time 'logistics' contains the time for loading respectively unloading the trucks as well as time for the outward and return journey between the filling station respectively storage and the site of operation, which is tightly calculated as one

30    hour per truck. It is assumed that there is an unrestricted amount of trucks available, which is of course a theoretical value, resulting in an overall time for logistics of one hour. In reality, the overall time would increase depending on the actual available amount of trucks. Time 'installation' refers to the installation of the specific system, if necessary, including additional time for a sandbag chain. According to this, time 'dismantling' refers to the dismantling of the individual systems as well as time for

cleaning of the SBRSs, if necessary also including additional time for a sandbag chain. Time for disposal or stowage of SBRSs was not taken into account.

The time, materials and helper advantages of the SBRSs are clearly visible. In case 1, the use of SBRSs requires approx. 25 %-30 % of the time, approx. 5 %-7 % of the helper hours and approx. 5 % of the trucks compared to the sandbag dam. If more helpers or trucks are used, the respective proportions shift, but the total effort remains the same. In case 2 and case 3, approx. 40 % of the time and approx. 6 % of the helper hours are required when using SBRSs. The logistics data in case 2 and case 3 were rounded up to fully loaded trucks. Eight Flutschutz-load drains or Flutschutz-ring dams can be transported per truck, so that when using these SBRSs there is a need for only approx. 8 %-9 % of the trucks required for sandbagging.

When sandbagging is used, poor access — and thus the need for sandbags being passed on over longer distances by means of a sandbag chain (see Figure 2) — may result in a significant additional need for helpers or the use of alternative means of transport such as helicopters or boats, which can only transport sandbags in small amounts This can also considerably increase the time required for transport as well as the costs incurred. The possible scenarios are manifold and could therefore not be considered in detail. SBRSs do not need additional helpers in case of poor accessibility, because due to their relatively low weight they can be put in place with much more easy means in the required amount, e.g. by the use of special vehicles which can access even wet ground but which cannot carry a lot of weight.

**Table 7: Comparison of time, helpers and logistics requirements for the installation and dismantling of sandbag and sandbag replacement systems - temporary flood dam (case 1).**

|  | Sandbag dam | Flutschutz-DCT | aqua defence | AQUARIWA |
|---|---|---|---|---|
| Helpers, incl. lower command | 10 | 2.5 | 2.5 | 2.5 |
| Trucks | 26 | 2 | 2 | 1 |
| **Installation** | | | | |
| Time materials [h] | 41.25 | - | 2.00 | 2.00 |
| Time logistics [h] | 1.00 | 1.00 | 1.00 | 1.00 |
| Time installation [h] | 20.63 | 7.50 | 6.48 | 8.71 |
| Total time, incl. logistics [h] | 62.88 | 8.50 | 9.48 | 11.71 |
| Total helper hours [h] | 618.75 | 18.75 | 21.20 | 26.78 |
| **Dismantling** | | | | |
| Time materials [h] | - | - | - | - |
| Time logistics | 1.00 | 1.00 | 1.00 | 1.00 |
| Time dismantling, incl. cleaning SBRS [h] | 20.63 | 16.50 | 12.96 | 7.10 |
| Total time, incl. logistics [h] | 21.63 | 17.50 | 13.96 | 8.10 |
| Total helper hours [h] | 206.25 | 41.25 | 32.40 | 17.76 |
| **Installation and dismantling** | | | | |
| Total time, incl. logistics [h] | 84.50 | 26.00 | 23.44 | 19.81 |
| Total helper hours [h] | 825.00 | 60.00 | 53.60 | 44.53 |

**Table 8: Comparison of time, helpers and logistics requirements for the installation and dismantling of sandbag and sandbag replacement systems – load drain (case 2) and ring dam (case 3).**

| | Load drain | | Ring dam | |
| --- | --- | --- | --- | --- |
| | **Sandbag** | **Flutschutz** | **Sandbag** | **Flutschutz** |
| Helpers, incl. lower command | 10 | 2.5 | 10 | 2.5 |
| Trucks | 2 | 1 | 2 | 1 |
| **Installation** | | | | |
| Time materials [h] | 2.45 | - | 2.25 | - |
| Time logistics [h] | 1.00 | 1.00 | 1.00 | 1.00 |
| Time installation [h] | 2.45 | 0.50 | 2.25 | 0.50 |
| Total time, incl. logistics [h] | 5.90 | 1.50 | 5.50 | 1.50 |
| Total helper hours [h] | 49.00 | 1.25 | 45.00 | 1.25 |
| **Dismantling** | | | | |
| Time materials [h] | - | - | - | - |
| Time logistics | 1.00 | 1.00 | 1.00 | 1.00 |
| Time dismantling, incl. cleaning SBRS [h] | 2.45 | 1.10 | 2.25 | 1.10 |
| Total time incl. logistics [h] | 3.45 | 2.10 | 3.25 | 2.10 |
| Total helper hours [h] | 24.50 | 2.75 | 22.50 | 2.75 |
| **Installation and dismantling** | | | | |
| Total time, incl. logistics [h] | 9.35 | 3.60 | 8.75 | 3.60 |
| Total helper hours [h] | 73.50 | 4.00 | 67.50 | 4.00 |

## 5 Conclusion

Tests of various SBRSs with the focus on stability, functionality and handling were carried out. The experiences from the test
setups show that SBRSs, owing to their functionality and their labour and time-saving characteristics as well as the lower
requirement for materials, offer the potential to make operational flood defence more efficient than with the use of sandbags
alone. Since SBRSs are technical systems whose functional capability must be proven before they can be used, the introduction
of a test and certification system is urgently recommended. A basis for the development of a certification system according to
the German standard is already available in the BWK leaflet 'Mobile Flood Protection Systems' (BWK, 2005), the international
certification systems such as FM Approvals (2019) or BSI Kitemark (2019a) as well as the test results described here and in
Massolle et al. (2018).

Further aspects have to be considered using SBRSs instead of sandbagging. These include the lower flexibility of SBRSs,
higher demands on trained personnel, the creation of hazards by assembly errors, defects in the construction, mechanical
influences due to flotsam, vehicles, and persons as well as by vandals, the possibility of collective failure (domino effect), or
the influences of currents, winds and waves. The hazards introduced through the use of SBRSs cannot entirely be ruled out,
but the hazard can be minimized by taking appropriate precautions, e.g. installing safety zones adjacent to the systems,
anchoring to the ground, tight monitoring of SBRSs and water side. Also SBRSs easily allow to impound higher flood water

levels, which is on the one hand an advantage but on the other hand results in greater probability of subsoil failure if high water levels are impounded. In general, the use of SBRSs can lead to higher requirements on a suitable subsoil. Many of the aspects mentioned can be laid down in guidelines to support decision-makers with regard to the possible use of SBRSs. However, taking into account possible catastrophic consequences in the event of failure, the installation of SBRSs should be planned and

5 executed under the supervision of specialists and under special observation during the flood event. From the authors' point of view SBRSs are rather a suitable supplement to than a full replacement of sandbagging. Especially because of their easy, flexible handling and their reliable usability within the scope of its possibilities, sandbags are an essential mean in the operational flood defence. No matter whether SBRSs find increasing application in future, sandbags will continue to play an important role in flood defence owing to their simple application and high flexibility — even if, for example, they are only

10 used to close gaps for which prefabricated systems of a certain length are not suitable.

The authors' determination of the operational costs was carried out for specific scenarios and with several simplifications, but nevertheless allows an approximate estimate of the operational costs of sandbagging and SBRSs under realistic conditions. When used once, all SBRS show higher overall costs, including costs for investment, logistics, installation and dismantling. The higher total costs result from the higher acquisition costs of the investigated SBRSs. SBRSs are reusable, therefore, with

15 regard to amortization of the higher acquisition costs of SBRSs, the number of times a system can be used within its service life plays a decisive role, since the acquisition costs of the investigated systems are amortized during their subsequent reuse. Since SBRSs can be transported with comparatively low logistical effort, a more centralised storage system is conceivable, so that in the event of flooding, the systems can also be transported from more distant regions that are not immediately affected by the flood. This would be in the interest of a cross-municipal and therefore cost-effective acquisition.

All investigated SBRSs show clear time-, material- and personnel-saving advantages. In particular, the time but also material and personnel saved during operation must be taken into account here, which may even be crucial to providing protection in the first place. The time, material and personnel saving characteristic of SBRSs might offer the possibility to use SBRSs during heavy precipitation events, respectively flood events with only short early warning times. Such events can come along with high flow velocities, resulting in high potential dynamic loads. Further investigations and a special testing routine would be

necessary in order to make reliable statements about a SBRSs' functionalities during such events.

From a technical point of view decision makers are confronted with the question of the reliability of SBRSs, which in general show a good functionality comparable to sandbagging, and in terms of time, personnel and material need show better results than sandbagging alone. The question of the functionality of SBRSs can be addressed by introducing independent test routines and certifications. From an economic point of view, decision makers are confronted with the question of higher investment

costs if SBRSs are purchased. The here done investigations indicate that only if SBRSs are subsequently reused, this is not connected to economic losses. In addition to the economic aspects, however, it should also be noted that SBRSs can be set up in a significantly shorter time, which often can be the basis for effective protection.

# Appendix

**Table A1: System dimensions and further properties of tested SBRSs.**

| Manufacturer/ distributor | Product name | Water height [m] | System height [m] | Length [m] | Width [m] | Weight (unfilled) [kg] | Diameter [m] | Main material | Fill material | Water permeable | Anchoring | Material requirements | Homepage |
|---|---|---|---|---|---|---|---|---|---|---|---|---|---|
| **Basin system** | | | | | | | | | | | | | |
| Aquariwa GmbH | AQUARIWA | 0,5-1,0 (water filled) 1,0-1,5 (sand filled) | 0,9-1,5 | - | - | 15,0-39,0 | 1 - 1,5 | Glass fibre-reinforced board with grid, foil water sack | Water, sand, gravel | No | None | Tarpaulins, sandbags, hoses, pumps or wheel loaders, dumpers | http://www.aquariwa.de/home/ |
| Indutainer | INDUTAINER | N.s. | 1,05 | 0,93 | 0,93 | 7,0 | - | Polypropylene fabric, polyurethane foam | Water | No | None | Tarpaulin, pumps, hoses, if necessary filling aid | http://www.indutainer.com/ |
| Quick Damm GmbH | Quick Damm | 1,0 | 1,0 | 2,0 | 1,0 | 50,0 | - | Steel frame, geotextile or plastic-coated fabric tarpaulin | Water, sand, gravel | No / Yes | None | Wheel loaders, dumper or pumps, hoses | http://www.quick-damm.de/start.html |
| **Trestle system** | | | | | | | | | | | | | |
| ALTRAD Plettac assco GmbH | aqua defence | 1,3 | 1,3 | 1,3 | 1,71 | app. 41,5 per m | - | Hard foam panels, galvanized support elements, tarpaulin | - | No | N.s. | Sandbags | https://plettac-assco.de |
| RS Stepanek KG | Aqua Barrier | 0,65-1,8 | 0,65-1,8 | 1,23 | N.s. | N.s. | - | Wooden pallets, galvanised support elements | - | No | N.s. | Wooden pallets, tarpaulins, sandbags | http://hochWaterschutz-rs.de/a/hochWaterschutz/aqua-barrier/ |
| **Dam system** | | | | | | | | | | | | | |
| NOAQ Flood Protection | NOAQ Boxwall | 0,5 | 0,528 | 0,705 | 0,68 | 3,4 | - | ABS (acrylonitrile butadiene styrene) | - | No | None | None | http://noaq.com/de/home-2/ |
| **Tube system** | | | | | | | | | | | | | |
| European Flood Control GmbH | Tiger Dam | 0,4;0,75;1,0 | 0,5 per tube | 15,0 | - | 30 per tube | 0,5 per tube | PVC | Water | No | For additional protection against slipping possible | Tarpaulins, sandbags, pumps, hoses | http://www.eu-floodcontrol.eu/ |
| Hochwasserschutz Agentur | Hydrobaffle | 0,23-1,83 | 0,31-2,44 | 3,0-32,0 | 0,7-5,49 | 2,61-20,32 | 0,7-5,49 | Plastic-coated fabric tarpaulin | Water | No | None | Pumps, hoses, square wrenches for the seals | http://www.hochwasserschutz-agentur.de/de/mobiler-hochWaterschutz-details |
| Mobildeich GmbH | Mobildeich | 0,45-2,6 | 0,45-2,6 | 10,0-50,0 | 0,4-2,6 | 5-56 per m | 0,45-1,5 | PVC coated polyester fabric | Water | No | None | Pumps, hoses, Y-piece | http://www.mobildeich.de/de/index.php |
| optimal Planen- & Umwelttechnik GmbH | Flutschutz-Doppelkammerschlauch | 0,6 | 0,9 | 10,0;15,0;20,0 | - | 110,0-220,0 | 1,5 / 0,9 | Polyester fabric coated on both sides | Water | No | None | Leaf blower, pumps, hoses | http://www.alles-optimal.de/index.php/produkte-fuer-feuerwehr-und-katastrophenschutz/hochWaterschutz/doppelkammerschlauch-flutschutz/75-doppelkammerschlauch-flutschutz |
| Öko-Tec Umweltschutz-systeme GmbH | Öko-Tec Schlauchwall | 0,5-1,5 | 0,5-1,5 | 5,0-20,0 | 1,5-2,3 | 4,0-9,5 per m | 0,5-1,5 | PVC | Air | No | If necessary, with wind on the side of the protection zone or with current on the water side | Blower | http://noaq.com/de/home-2/ |
| **Load drain** | | | | | | | | | | | | | |
| optimal Planen- & Umwelttechnik GmbH | Flutschutz-Auflast | - | 0,6 | 7,0 | 3,5 | 70,0 | - | Polyester fabric coated on both sides | Water | No | None | Pumps, hoses | http://www.alles-optimal.de/index.php/produkte-fuer-feuerwehr-und-katastrophenschutz/hochWaterschutz/schlauchlastfilter-flutschutz/77-schlauchlastfilter |
| **Temporary ring dam** | | | | | | | | | | | | | |
| optimal Planen- & Umwelttechnik GmbH | Flutschutz-Quellkade | 1,0 | 1,0 | 2,9 | 2,9 | 35,0 | - | Polyester fabric coated on both sides | Water | No | None | Leaf blower, pumps, hoses | http://www.alles-optimal.de/index.php/produkte-fuer-feuerwehr-und-katastrophenschutz/hochWaterschutz/schlauchkade-flutschutz/76-schlauchkade-flutschutz |

## Author contribution

5 Conceptualisation: B.K.; Methodology: L.L. and C.M.; Resources: C.M. and L.L.; Formal Analysis: L.L., C.M. and V.K.; Writing—original draft preparation: L.L.; Writing—review and editing: B.K., C.M. and L.L.; Visualisation: L.L. and V.K.; Supervision: B.K. and L.L.; Project administration: B.K.; Funding acquisition: B.K.

**Conflicts of interest**

The authors declare that they have no conflict of interest.

**Acknowledgements**

The test setups were carried out within the framework of the project 'Adaptation of flood protection training and dyke defence
of the THW Training Centre Hoya to the challenges of climate change' (HWS-Bildung, duration 2016 - 2018), funded within
the framework of the German Adaptation Strategy to Climate Change by the Federal Ministry for the Environment, Nature
Conservation and Nuclear Safety and by the THW Foundation. We would like to thank the manufacturers and their distributors
for making their systems available for the tests and our student assistants for their active support during the test setups. We
would furthermore like to thank the anonymous referees for their helpful comments and suggestions.

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
