# Peer review of "Sandbag Replacement Systems - a nonsensical and costly alternative to sandbagging?"

_Natural Hazards and Earth System Sciences, 2019_

## Referee Comment (RC1) · Anonymous Referee #1 · 5 Jul 2019

The manuscript compares the use of sandbags with alternative flood protection systems, the so-called Sandbag Replacement Systems (SBRS), in terms of costs, helpers and logistics for installation and dismantling. Three different cases are considered: temporary flood dam, load drain in the case of a saturated dyke over an extensive area and ring dike for reinforcement against heavy punctual exit of seepage on the inner embankment of the dyke. The manuscript is well written and structured and addresses, despite a lot of simplifications, an interesting and rarely explored topic in the literature, that is the assessment of alternative protection systems in case of a inundation event, which can be more efficient and convenient than the traditional sandbags. Although this, the paper does not address the topic of the efficiency of these methods and it is limited to cite the other manuscript of the same authors (and, for example, not in

the introduction, where it is fundamental in order to undertand why the authors write about this topic), in which the topic is discussed. It is not clear, in the current version of the manuscript, why it is so important to focus on the SBRS and the reason why a comparison in terms of costs, helpers and logistic is necessary. The authors in this manuscript compared three different type of SBRS, but it is not clear why they take these methods among all the availables one (the reason is maybe that they have better performances, but it can only understood by reading the other manuscript). In addition, the SBRS are very shortly described, taking for granted that their characteristics are clear and well known (but it is not always obvious). In my opinion, the manuscript needs to be re-structured considering also a part about the "hydraulic" efficiency of the SBRS, because, in the current version, it is not enough interesting to be published in NHESS, but can be important more for municipal administrations. The manuscript would earn a lot in terms of quality with the consideration of the "hydraulic tests" of the other manuscript. I suggest to the authors to consider the idea of unifying the two manuscripts, which seem to be a bit poor if considered singularly.

In case authors consider to review the manuscript in this direction, I would also suggest to better justify some values, for example the costs of trucks and fuel (it is not clear to me which is the considered distance to cover), the sandbag requirement (Table 1: why Acqua defence and Aquariva need sand?) and the price used for sand (it is written, during the event it rises sharply: which price is considered in the manuscript? during the event? an average value? etc.) I would also better explain what the terms in the tables mean, for example "time materials", "time logistics", etc. (Table 3, e.g.) In Sec. 3.2, it is not clear to me why SBRS don't require additional helpers in case of poor acces to the site, as it happens for sandbags. In the conclusions, I couldn't find anything about the higher costs of SBRS, although I think it is a relevant result of the analysis. I think some considerations on the long terms is also needed, in order to say that the higher costs of SBRS are amortized becaause they can be reused. I was also curious to know if there are studies on the case in which these protection systems turn out to be undersized and are, for example, overtopped: can they be reused? how is

the amortization reduced?

Finally, I found a couple of other papers which addressed the comparison of the performance of different flood protection systems, and which the authors can consider as additional material: Wibowo & Ward, 2016 "Evaluation of temporary flood-fighting structures" and Rappazzo & Aronica, 2016 "Effectiveness and applicability of flood barriers for risk mitigation in flash-flood prone Mediterranean area".

---

## Referee Comment (RC2) · Anonymous Referee #2 · 16 Jul 2019

The article describes Sandbag Replacement Systems (SBRS) in relation to sandbags during emergency management of floods and is interesting to read. It contains research results in terms of costs, helpers and logistics. I recommend a review (major revisions) and a focus on the following aspects:

- The article is very much orientated to the situation in Germany. Existing literature comparing different flood protection systems is not used in a sufficient way. Especially the experience of the emergency management during the recent flood disasters in Europe should be considered. In addition to that, a proper explanation (e. g. figure 1) and a translation of the German terms are necessary.
- It is not quite clear which applications of SBRS and sandbags are considered. In chapter 2, "extreme flood events" are mentioned for the technical assistance of THW. But is SBRS really a tool for "extreme events"?
- The authors compare SBRS with the use of sandbags but do not describe the hydraulics. In my opinion, without a substantial description of the hydraulic situation this is difficult to understand. In Figure 1, "temporary flood protection dams" are visualized. But there is no explanation, if all the four SBRS are equal to each other and what their behavior will be as a response to different hydraulic conditions.
- The overall description of SBRS should take into consideration relevant technical aspects. In chapter 2, some statements are made, e. g. relating "mechanical influences" or "vandalism" without a detailed discussion.
- Chapter 3 mainly focuses on costs. It is somehow confusing to understand the exact definition for example of "helpers". Some other terms are used in the article: "employee", "THW-helper" etc. Is this all the same? Is it used as a synonym? Are the costs always the same, no matter if a helper has a special expertise and training (like the "THW helper")? And is 5 EURO per helper and day (chapter 3.2) really a serious value?
- Some of the mentioned costs (table 1 and 2) need clarification: What does "cost of materials" or "cost of trucks" mean? In the text, it is expressed that there are "comparatively low costs for the use of THW vehicles". But is this for Germany only?! Then it should be clarified that all the statements are based on figures and numbers of the German disaster relief system.
- Table 3: For SBRS systems "2,5 helpers including lower command" are needed? What does that mean?
- The conclusion is not based on the research results. A number of "political statements" are mentioned, e. g. "In particular it is necessary to provide financial support to the municipal authorities responsible for the purchase of the system". No arguments for this statement are mentioned or discussed throughout the article. Overall, I find the conclusion more relevant for stakeholders or municipalities but with little scientific value.
- I recommend to review the English language and strongly emphasise a correct use of technical terms, e. g. emergency or disaster management.

---

## Author Comment (AC1) · 21 Aug 2019

Dear Referee #1, Thank you very much for your review. Your proposal to link the two publications nhess-2019-164 and nhess-2019-165 is gladly accepted. This could provide a more understandable introduction to the problem without the risk of repetition of passages. Concerning the hydraulic efficiency, we would like to add a summary paragraph on water heads and seepage rates realized in testing but would like to refer to the paper Massolle et al. 2018 for more information. Considering justification of values like costs of trucks and fuel we will add more information on e.g. calculated distances. We will review all positions for traceability. Furthermore, we will state clearly the meaning of terms in the tables like 'time logistics' etc. Additionally, we will explain the chosen numbers e.g. for the need of helpers for specific access conditions etc. In the conclu-

sions, we will include a paragraph on investment costs for both, sandbag systems and SBRS, to complete the statements on the cost calculations. In addition, the cost effect of multiple use of SBRS will be discussed and the topic repair possibility of SBRS will be included in the paper. The Authors are grateful for naming the additional literature. The paper by Wibowo & Ward (2016) is based on the tests published by Pinkard et al. (2007), which is cited in the manuscript. Rappazzo & Aronica (2016) focus mainly on pre-installed SBRS, which are outside the scope of the investigations. In the final version we will check whether we can consider the named papers as well as others for a broader view on the topic.

---

## Author Comment (AC2) · 21 Aug 2019

Dear Referee #2, Thank you very much for your review. At the suggestion of two further reviews on this article (nhess-2019-165) and on the article nhess-2019-164 it is planned to summarize the two articles. This will also include more literary references not only from Germany, but also beyond. We will include a proper explanation to Figure 1. Considering the addressed German terms in Table 1 – these terms are product names. Nevertheless, we will try to translate them into English, where possible and appropriate. Considering the mentioned application field of SBRS: Sandbags as well as SBRS are used in emergency flood control – especially in case permanent flood protection systems like dikes are failing or in case no permanent flood protections schemes are available because the currently endangered area was thought not

to be at risk. Thus, sandbags as well as SBRS are certainly used in extreme flood events. Concerning the hydraulic efficiency of sandbags and SBRS, we would like to add a summary paragraph on water heads and seepage rates realized in testing but would also like to refer to the paper Massolle et al. 2018 for more information. In Massolle et al. 2018, the hydraulic testing of different sandbag and SBRS systems also relevant for the present papers nhess-2019-164 and nhess-2019-165 are described in detail. By merging the two papers nhess-2019-164 and nhess-2019-165 discussion on 'mechanical influences' and 'vandalism' will be added. In chapter 3 the terms 'helpers', 'THW-helpers', 'employees' will be checked for consistency. Furthermore, a clear explanation will be given on cost calculation for helpers (e.g. costs for trained or not-trained helpers). In chapter 3.1, the cost estimation of 5 Euro per helper and day is only related to the costs for upper control like disaster control management, technical incident command and platoon, but not for the helper itself. These are realistic overhead costs related to the number of helpers in action. Considering justification of values like 'costs of materials' and 'costs of trucks' etc. we will add more information and will review all positions for traceability. In Table 3 the term '2.5 helpers including lower command' means that 2 helpers are busy in installation and dismantling of the SBRS. Another one helper on the lower command level can take care for two different areas of application – therefore only half a helper is counted. In total 2.5 helpers. Concerning the conclusion: we will clean the conclusion from political statements. Concerning the English language – we will have another review of the grammar and in particular will check the technical terms for accuracy and consistency.

---

## Author Response (AR2)

**Authors' reply to:**

Referee #1, Report #2:

The manuscript discusses advantages and disadvantages of alternative flood protection systems to sandbags, the so-called Sandbag Replacement Systems (SBRS), taking into consideration their functionality, stability and handling. Sandbags or SBRS are also compared in terms of costs, helpers and logistics for installation and dismantling in three different cases:
temporary flood dam, load drain in the case of a saturated dyke over an extensive area
and ring dike for reinforcement against heavy punctual exit of seepage on the inner embankment of the dyke.
The manuscript is well written and structured and it improved a lot since the previous version, mainly due to the merging of the two previous manuscripts and the clarification of many concepts.
Although I think that it doesn't add so much in terms of scientific knowledges (especially the first part that is little more than a summary of Masolle et al., 2018), it addresses an interesting and rarely explored topic in the literature, and it can be interesting for decision makers and stakeholders, in order to improve the management of extreme events.
I will suggest to the author some modification more, in order to improve the scientific side even more and make the manuscript more suitable to be published in NHESS (I will refer to pages and lines of the version without track changes).

- Concerning the introduction, I would try to be more clear in saying exactly what is the state of the art, which are the gaps and which are the aims of the study.

  **Reply:** Please, especially refer to P7L12 et seqq (version without track changes).

- For example, at P2L7-8, authors say "SBRSs are suitable for flood protection and can be equated with sandbagging in terms of functionality"... It seems to be a conclusion based on the results of the study, it is not clear what author refer (if necessary, add citations).

  **Reply:** A citation has been added.

- Similarly, there are other sentences that seem to be already a conclusion, while they should present the topic to be assessed in the study (e. g. P3L8-10, or consideration about costs and functionality.

  **Reply:**
  - P3L8-10 – has been revised.
  - Some of the sentences, which seem to be a conclusion, can already be derived from general reflections on the subject. These sentences help to introduce the topic, why more in-depth investigations on SBRSs are important.

- I think the reference to Fig. 7 is too far in the Introduction (P2L9), please revise this part.

  **Reply:** The reference to Fig. 7 in the introduction has been revised.

- Statements at P3L13-14 and P4L16-17 need references. P5L11: ")." is missing, after 2019b. Please revise also sentence at P6L12-13.

  **Reply:** The sentences have been revised.

- I strongly suggest to add, before actual chapter 2, a chapter with the description of all flood defense methods considered in the study, included a brief description of sandbags and their

construction, installation and possible reuse. This is useful, in my opinion, to better merge the two parts coming from the two previous manuscripts.

**Reply:** A new chapter 2 has been added.

− My main concern is about actual chapter 2: after having described a lot of SBRSs, tests are performed just on a few of these (or they are only partially performed) because of different reasons (accessibility conditions, avoidance of damages, difficulties in sand fillings, unevenness of the basin floor, particular condition of the floor with consequent neglection of underground failure risk, short duration of the tests, to mention some of them). In this case, are the results really valid? Can we talk about functionality, stabilty and handling of SBRSs in general? I think some clarification from the beginning are needed, in order not to expect results that are valid in general terms.

**Reply:** Corrections of the wording in the introduction and the title of the old chapter 2 has been done to be clearer on this aspect.

− In addition, I think a comparison with the performance of sandbags could give more coherence with the title and the rest of the manuscript (at the moment, functionality is a comparison term for only the different type of SBRSs, while costs and logistic are compared also with sandbags).

**Reply:** A comparing part also for the performance of sandbags has been added in table 3.

− Concerning the results part, I would only focus on those results that can be quantify, in the scientific term. There are too many sentences that are too general and can be stated also before performing the tests (e.g. P12L8-11, P12L16-21, and others).

**Reply:** These statements can indeed be derived without performing the tests. However, during the test set-ups the importance of these statements also became clear. Furthermore, because the results are important for decision-makers and practitioners the authors believe that it is important to collect in this paper even such rather simple test results, in order to get a comprehensive overview of the aspects concerning the use of SBRSs.

− I would better describe Fig. 10, included what the red line means.

**Reply:** An explanation has been added in the labelling of figure 10.

− In Table 1, I wonder how can authors state results about wind influence or similar, when at P8L10-11, it is written that current, waves, wind flotsam and vessel impact cannot be investigated in the test facility.

**Reply:** The results are stated on general considerations, which can be derived from the system properties (great weight).

− P15L19: please specify (with examples) what "authors' consideration" means.

**Reply:** Authors' consideration means basically theoretical assessments, based on the experiences of the tests or general assumptions, thus resulting in own estimates. For example the possible occurrence of a domino effect can be derived from theoretical reflections on the set-up of a SBRS.

– Chapter 3: I would shorten this chapter a bit, also moving some part in the new descriptive chapter, e.g. the duration of sandbags and SBRSs (P20L19-29)

**Reply:** The old chapter 3 has been revised and the corresponding statements have been moved to the new chapter 2.

– Conclusion: in general, I would avoid numbers, costs and percentages in this chapter, moving them in the results' chapter. P25L17: please specify the precautions you cite. P25L17-P26L2: how can you speak about subsoil failure risk if you didn't consider it in the tests?

**Reply:**
– Statements about numbers, costs and percentages have been removed
– P25L17: Examples have been added
– P25L17-P26L2: It is known – not least by own experience during flood events - that subsoil failure risks dependent on the hydraulic gradient, which is in principle increasing with increasing water levels.

[revised manuscript text omitted]